# Rab41-mediated ESCRT machinery repairs membrane rupture by a bacterial toxin in xenophagy

Takashi Nozawa [1], Hirotaka Toh[1], Junpei Iibushi[1], Kohei Kogai[1], Atsuko Minowa-Nozawa[1], Junko Satoh[2], Shinji Ito[2], Kazunori Murase[1] & Ichiro Nakagawa [1] ✉

Xenophagy, a type of selective autophagy, is a bactericidal membrane trafficking that targets cytosolic bacterial pathogens, but the membrane homeostatic system to cope with bacterial infection in xenophagy is not known. Here, we show that the endosomal sorting complexes required for transport (ESCRT) machinery is needed to maintain homeostasis of xenophagolysosomes damaged by a bacterial toxin, which is regulated through the TOM1L2–Rab41 pathway that recruits AAA-ATPase VPS4. We screened Rab GTPases and identified Rab41 as critical for maintaining the acidification of xenophagolysosomes. Confocal microscopy revealed that ESCRT components were recruited to the entire xenophagolysosome, and this recruitment was inhibited by intrabody expression against bacterial cytolysin, indicating that ESCRT targets xenophagolysosomes in response to a bacterial toxin. Rab41 translocates to damaged autophagic membranes via adaptor protein TOM1L2 and recruits VPS4 to complete ESCRT-mediated membrane repair in a unique GTPase-independent manner. Finally, we demonstrate that the TOM1L2–Rab41 pathway-mediated ESCRT is critical for the efficient clearance of bacteria through xenophagy.

Almost all intracellular bacteria invade host cells through endocytic membrane trafficking. Because bacteria that invade through endocytosis are rapidly transported to degradative organelles, such as lysosomes, bacterial pathogens have evolved strategies to modulate intracellular bactericidal pathways during infection for survival and proliferation. Some bacterial pathogens, such as *Shigella flexineri* and *Salmonella enterica* Typhimurium, translocate virulence factors to interfere with regulators of host membrane trafficking and avoid fusion with lysosomes to establish a replicative compartment[1–3]. Alternatively, *Staphylococcus aureus, Listeria monocytogenes*, and Group A *Streptococcus* (GAS) secrete cytolysins to damage and rupture the vacuole membrane and invade the host cytoplasm[4–6]. To prevent the leakage of pathogen virulence and bacterial cytosolic

invasion, host cells have two steps of mechanisms; one is endosomal/lysosomal membrane repair by the ESCRT (endosomal sorting complexes required for transport) machinery[7–10]. When this fails, selective autophagy (also termed xenophagy) targets cytosolic bacteria and transports them to lysosomes (xenophagolysosomes) for degradation[6,11,12].

ESCRT proteins were initially identified as regulators of ubiquitinated cargo sorting into multivesicular bodies but now have been established to have various intracellular membrane dynamics processes such as budding of enveloped viruses, vesicle budding, cytokinetic abscission, nuclear envelope maintenance, repair of the plasma membrane and endolysosomal membrane, and autophagy[13,14]. The ESCRT machinery can be divided into four functionally distinct

[1]Department of Microbiology, Graduate School of Medicine, Kyoto University, Yoshida-Konoe-cho, Sakyo-ku, Kyoto 606-8501, Japan. [2]Medical Research Support Center, Graduate School of Medicine, Kyoto University, Yoshida-Konoe-cho, Sakyo-ku, Kyoto 606-8501, Japan. ✉e-mail: nakagawa.ichiro.7w@kyoto-u.ac.jp

complexes: ESCRT- 0, ESCRT- I, ESCRT-II, and ESCRT-III. ESCRT-III, which consists of α-helical CHMP proteins, is recruited by ESCRT-II and critical for driving membrane constriction through the formation of membrane-binding spirals that mediate membrane deformation and scission, in cooperation with the ATPase VPS4[13–16]. ESCRT-III is reported to be recruited to bacteria-containing phagolysosomes to repair membrane damage by bacterial factors or membrane tension caused by bacterial proliferation[7,10,17–20]. During *Mycobacteria* infection, inhibition of ESCRT proteins resulted in bacteria accessing the cytosol prematurely, indicating that ESCRT maintains bacteria-containing phagosomal integrity[7]. ESCRT recruitment to the endolysosomal membrane is trigged by $Ca^{2+}$ release and $Ca^{2+}$-responding proteins as well as during plasma membrane repair[9,21–23], and a β-galactoside-binding cytosolic lectin Galectin-3 is also reported to control ESCRT-III recruitment[8]. In addition, recent reports showed that the kinase leucine-rich repeat kinase 2 (LRRK2) and Rab8A axis are involved in ESCRT-III recruitment[10].

Xenophagy is activated in response to extensive endolysosomal membrane damage or bacterial escape into the cytosol. Cytosolic bacteria is marked ubiquitin or other recognizing proteins, selectively targeted by xenophagic membrane and surrounded by xenophagosomes[12,24]. The bacteria-containing xenophagosome fuses with a lysosome to form a degradative xenophagolysosome, and inside bacteria is thought to be efficiently degraded. Thus, despite various bacterial pathogens rupturing endosomal/phagosomal vacuoles, whether xenophagosomal vacuoles are also damaged by bacteria is unknown.

Rab guanosine triphosphatases (Rab GTPases) are key regulators of intracellular membrane trafficking, and more than 60 Rabs have been identified in human cells[25,26]. Rab proteins function as molecular switches that alternate between the active GTP-bound and inactive GDP-bound forms, as coordinated by specific GTPase-activating proteins and GTPase exchange factors. Activated Rab interacts with various effectors to elicit distinct downstream events. Rab proteins regulate starvation-induced autophagy and selective autophagy, including xenophagy[27,28]. For example, Rab1A regulates ULK1 trafficking through effector C9orf72[29], and the Rab1A-dependent pathway is critical for xenophagy of *Salmonella*[30]. Mouse Rab41, which is a homolog of human Rab40C, is required for xenophagy of *Streptococcus pneumoniae* via a GTPase-dependent mechanism[31]. We also reported that endosome-resident Rab35 recruits autophagy receptor NDP52 as a target in xenophagy[32], and Rab9A, Rab17, and Rab30 are involved in xenophagosome formation or its lysosomal fusion as a molecular switch[33–36]. Thus, GTPase-dependent Rab proteins have been shown to regulate the various steps of xenophagy, such as bacterial recognition, the formation of xenophagosomes and their initiation signals, and the lysosome fusion process. However, a comprehensive analysis of Rab proteins in xenophagy has not been performed, and thus we are still far from a complete understanding of xenophagic membrane regulation through Rab proteins.

In this study, we performed comprehensive siRNA-screening and identified Rab41, which is critical to maintain the homeostasis of xenophagolysosomes damaged by bacterial toxins. We demonstrate that GTPase-independent Rab41 and TOM1L2 regulate VPS4 dynamics to control endosomal sorting complexes required for transport (ESCRT)-mediated repair of the damaged membrane.

## Results

### Rab8A and Rab41 are required for xenophagy of GAS

For comprehensive screening of Rab GTPases involved in xenophagy against GAS infection, we first knocked down the expression of Rab GTPases in HeLa cells. We examined bacterial survival in cells at 6 h post-infection (hpi) compared with that at 2 hpi because intracellular GAS is degraded by xenophagy during this time[6]. Of the 61 Rabs tested,

knockdown of Rab2B, Rab3C, Rab5A, Rab7A, Rab8A, Rab9A, Rab11A, Rab17, Rab19, Rab20, Rab23, Rab25, Rab30, Rab35, Rab40B, Rab41, and Rab44 significantly increased bacterial survival (Fig. 1a), indicating that these Rab proteins are involved in bacterial clearance. Of these Rabs, Rab5A, Rab7A, Rab9A, Rab17, Rab23, Rab30, and Rab35 were previously reported to be involved in eliminating invading GAS through xenophagy or endosome-lysosome trafficking[32–35,37]. To determine whether Rab2B, Rab3C, Rab8A, Rab11A, Rab19, Rab25, Rab40B, Rab41, and Rab44 are involved in the elimination of the bacteria via xenophagy, we knocked down these Rabs in *ATG5*-knockout (KO) cells (i.e., xenophagy-deficient cells). We examined the viability of the bacteria in these cells. Knockdown of Rab2B, Rab3C, Rab11A, Rab20, Rab25, and Rab40B increased intracellular GAS even in xenophagy-deficient cells, whereas depletion of Rab8A, Rab19, Rab41, and Rab44 did not affect GAS survival, suggesting that Rab8A, Rab19, Rab41, and Rab44 are involved in the elimination of GAS through xenophagy (Fig. 1b). However, these data do not rule out the possibility that Rab2B, Rab3C, Rab11A, Rab20, Rab25, and Rab40B are involved in xenophagy.

To examine whether Rab8A, Rab19, Rab41, and Rab44 are involved in xenophagosome formation, we knocked down these Rabs in cells expressing GFP-LC3, an autophagic membrane marker (Supplementary Fig. 1a). We quantified the cells harboring GAS-containing LC3-autophagosome-like vacuoles (GcAVs) using confocal microscopy. The efficiency of GcAV formation significantly reduced in Rab19-knockdown cells (Fig. 1c, d). We also found that lysosomal fusion of GcAVs, LC3 vacuole signals overlapping lysosomal marker LAMP1, decreased in Rab19- and Rab44-knockdown cells (Fig. 1c, e). These findings suggested Rab19 is involved in GcAV formation and the lysosomal fusion process of GcAVs, and Rab44 is associated with GcAV–lysosome fusion.

We next monitored acidification within the xenophagolysosome during GAS infection using LysoTracker, an acidification indicator. The signal intensity of LysoTracker inside GcAVs increased with time in control cells (Fig. 1f, g). However, in Rab8A or Rab41 knockdown cells, the LysoTracker signal was transiently elevated from 2 hpi to 3 hpi and then decreased, suggesting that Rab8A and Rab41 are essential for xenophagolysosome membrane homeostasis. To validate the involvement of Rab8A and Rab41 in maintaining an acidic environment within xenophagosome and thereby promoting bactericidal activity, we generated *Rab8A* and *Rab41* knockout cells (Supplementary Fig. 1b) and infected these cells with GAS. Knockout of *Rab8A* or *Rab41* significantly decreased acidification of GcAVs at 4 hpi (Fig. 1h, i) and increased bacterial survival (Fig. 1j). Taken together, these data suggested that Rab8A and Rab41 are critical factors for xenophagy against GAS through maintaining GcAV membrane integrity.

### ESCRT targets GAS-containing autophagolysosomes through Rab8A and Rab41

Because Rab8A is reported to be involved in the repair of damaged endomembranes through the ESCRT-III machinery[10], we speculated that the xenophagolysosome membrane is damaged by bacteria and that membrane repair through ESCRT-III may be impaired in Rab8A-depleted cells. We then observed the localization of ESCRT-III component CHMP4B in GAS-infected cells and found that CHMP4B-EGFP was localized throughout the GcAV membrane at 4 hpi (Fig. 2a). Immunofluorescence also revealed that endogenous CHMP4B is localized to GcAVs (Fig. 2b). Moreover, not only ESCRT-III components, such as CHMP4B and CHMP6 but also VPS28 (ESCRT-I) and SNF8 (ESCRT-II) were recruited to GcAVs (Fig. 2c).

Damaged lysosomes are repaired by calcium release-dependent recruitment of the ESCRT machinery[22,23]. To examine whether ESCRT components are recruited to xenophagolysosomes in a calcium-

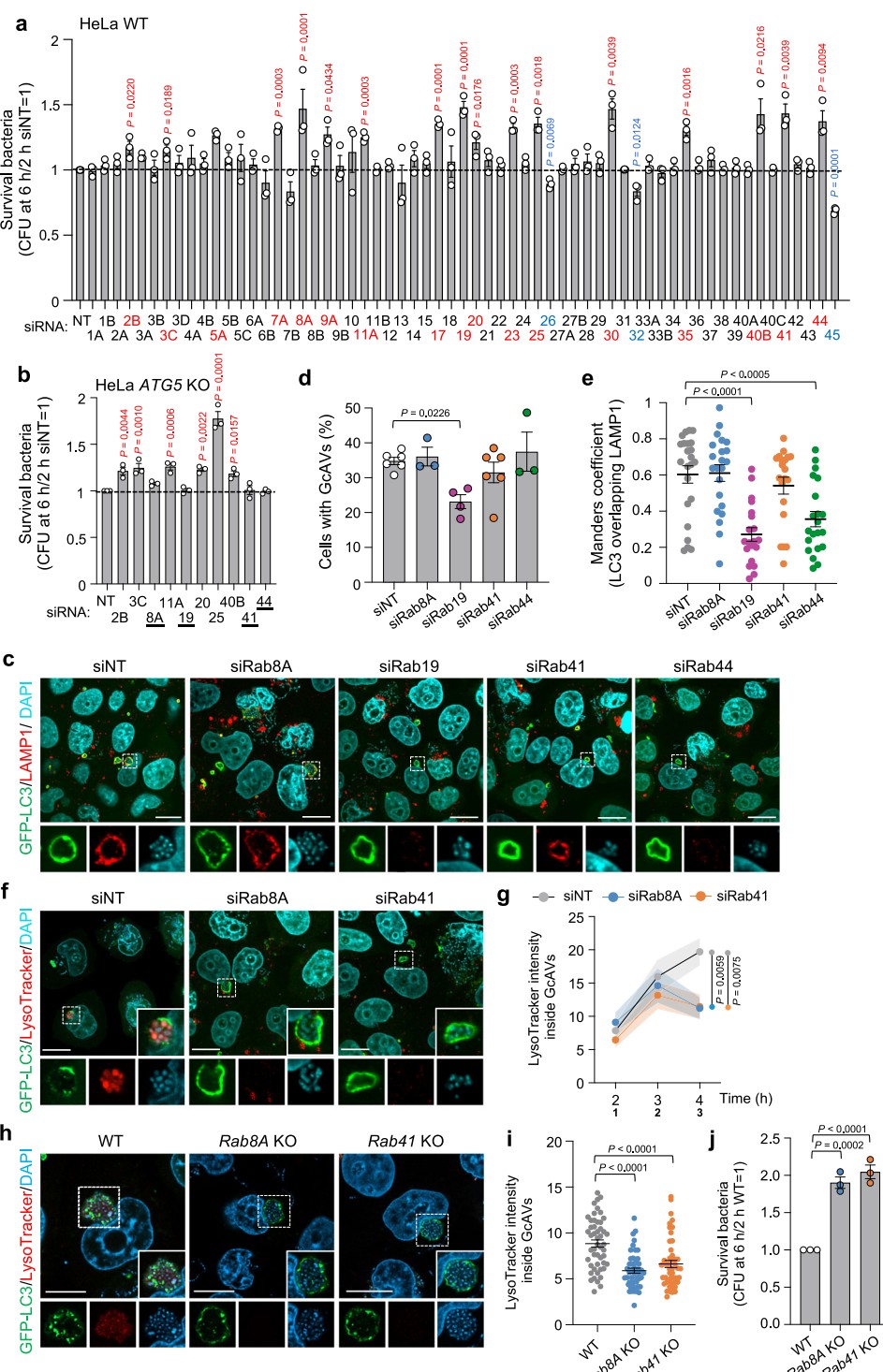

**Fig. 2 |** (panels a–j as shown)

dependent manner, we observed CHMP4B recruitment to GcAV in the presence of the calcium chelator BAPTA-AM. We found that BAPTA-AM treatment suppressed CHMP4B recruitment to GcAVs (Fig. 2d), indicating that a calcium-dependent mechanism induces the recruitment of ESCRT machinery to GcAVs.

To identify when ESCRT-III is recruited to GcAVs, we monitored the localization of GFP-LC3, CHMP4B, and LAMP1 at 2 to 6 hpi. LAMP1-positive GcAVs increased from 2 to 6 hpi, while CHMP4B-positive GcAVs increased from 3 to 4 hpi and then decreased (Fig. 2e, f). In addition, few LAMP1-negative/CHMP4B-positive GcAVs were observed, and LAMP1-positive/CHMP4B-positive GcAVs increased

over 3–4 hpi (Fig. 2g). These results suggested that ESCRT-III is recruited explicitly to xenophagolysosomes during GAS infection.

We next tested whether Rab8A and Rab41 are involved in recruiting ESCRT-III to GcAVs. Knockdown of Rab8A significantly decreased GcAVs overlapping CHMP4B, whereas knockdown of Rab41 increased CHMP4B signals on GcAVs (Fig. 2h, i). The measurement of CHMP4B recruitment to GcAVs showed that Rab8A-knockdown abolished CHMP4B recruitment, and CHMP4B was accumulated on GcAVs in Rab41-knockdown cells (Fig. 2j). Furthermore, knockout of *Rab8A* reduced CHMP4B signals on GcAVs, while that of *Rab41* increased (Fig. 2k, l). Thus, it was shown that

**Fig. 1 | siRNA screening reveals that Rab8A and Rab41 are critical to maintain the integrity of the autophagolysosomes of GAS. a, b** Screening for Rab GTPases that affect intracellular bacterial viability. HeLa wild-type (WT) (**a**) or *ATG5*-knock-out (KO) (**b**) cells transfected with the indicated siRNA were infected with GAS. siNT means non-targeting control siRNA. Colony counting (determination of colony forming unit, CFU) was conducted to determine the number of invading and surviving GAS; the bacterial survival data were calculated as the ratio of "intracellular live GAS at 6 h" to "total intracellular GAS at 2 h". Data are represented as individual values (circles) and mean ± standard error of the mean (SEM) (*n* = 3 biologically independent experiments). One-way analysis of variance (ANOVA), Dunnett's test. Red *P* values indicated a significant increase and blue ones means a significant decrease. **c–e** HeLa cells stably expressing GFP-LC3 were transfected with the indicated siRNA, and infected with GAS for 4 h, then fixed and immunostained for endogenous LAMP1 (red). Cellular and bacterial DNA were stained with DAPI (cyan). **c** Shown representative microscopic images are single slice. Scale bar, 10 μm. (**d**) GcAV formation was quantified as the percentages of cells with GcAVs. Data are represented as individual values and mean ± SEM of more than three independent experiments (200 <cells examined in each independent experiment). One-way ANOVA, Dunnett's test. **e** The proportion of the LC3 signal (GcAVs) that colocalized with LAMP1 was quantified by Mander's coefficient M1. Data are represented as individual values (circles) and mean ± SEM (30 < cells examined over three independent experiments). One-way ANOVA, Dunnett's test. **f, g** HeLa cells stably expressing GFP-LC3 were transfected with the indicated siRNA, then infected with GAS for the indicated time, and finally fixed. Cells were stained with LysoTracker Red 30 min prior to fixation. Shown confocal images are single slice (**f**) and the LysoTracker intensity inside GcAVs was quantified (20 < cells examined in each three independent experiments) (**g**). Scale bar, 10 μm. Data represent mean ± SEM. One-way ANOVA, Dunnett's test. **h, i** HeLa WT, *Rab8A* KO, *Rab41* KO cells were infected with GAS for 4 h, and finally fixed. Cells were stained with LysoTracker Red 30 min prior to fixation. Shown confocal images are single slice (**h**) and the Lyso-Tracker intensity inside GcAVs was quantified (**i**). Scale bar, 10 μm. Data are individual values (circles) and mean ± SEM (50 GcAVs examined over three independent experiments). One-way ANOVA, Dunnett's test. **J** HeLa WT, *Rab8A* KO, and *Rab41* KO cells were infected with GAS. CFU was determined to quantify the number of invading and surviving GAS at 2 h and 6 h after infection, respectively. Data are individual values (three independent experiments) and mean ± SEM. One-way ANOVA, Dunnett's test. Source data are provided as a Source Data file.

Rab8A and Rab41 regulate ESCRT-III recruitment and its dissociation process, respectively.

### GAS pore-forming toxin SLO triggers ESCRT-III recruitment and promotes intracellular survival

ESCRT-III recruitment to damaged lysosomes is triggered by membrane damage by bacteria or a membranolytic reagent[38]. In the case of GAS-infected cells, GAS secretes the pore-forming toxin SLO and damages the endosomal membrane[6], but it is not clear whether SLO also damages the xenophagolysosomal membrane. Because GAS mutants not expressing SLO does not induce xenophagy, we constructed an anti-SLO intracellular antibody (intrabody) to inhibit SLO secreted by cytosolic bacteria. We constructed a single-chain (sc) variable domain (Fc; termed scFv) derived from an anti-SLO monoclonal IgG (HS1) using a method designed to engineer an ultra-stable cytoplasmic antibody (STAND)[39]. It was shown that prophylactic and therapeutic injection of HS1 monoclonal IgG into mice significantly improved the survival rate following lethal infection with an invasive GAS isolate[40]. To check whether STAND-HS1 (the anti-SLO intrabody) binds to SLO secreted by GAS in cells, cells expressing STAND-HyHEL10 (control) or STAND-HS1 were infected with GAS, and STAND antibodies were precipitated using an anti-FLAG antibody. We found that SLO was precipitated with STAND-HS1, indicating that STAND-HS1 binds to SLO during GAS infection (Fig. 3a). We next investigated the effects of STAND-HS1 expression on the GAS infection process. Expression of STAND-HS1 did not affect bacterial invasion efficiency and xenophagy induction (Fig. 3b–d). However, ESCRT-III subunit CHMP4B recruitment to GcAVs significantly reduced, suggesting that the ESCRT machinery targets GcAVs in response to membrane damage by SLO (Fig. 3e, f). We further examined the acidification of these GcAVs that were not targeted by ESCRT and found that the intensity of LysoTracker in the GcAVs increased in response to the expression of STAND-HS1 (Fig. 3g). These results implied that SLO damages xenophagolysosomes and contributes to the inhibition of GcAV acidification. The ESCRT machinery targets GcAVs in response to pore formation by SLO. We also examined the effects of STAND-HS1 on intracellular bacterial survival. We found that the anti-SLO intrabody inhibited bacterial proliferation of wild-type GAS but not SLO-deficient GAS in cells (Fig. 3h), supporting the theory that some intracellular GAS escapes from xenophagolysosomes through SLO and facilitates bacterial survival in cells. It was suggested that cytosolic anti-SLO scFv does not target SLO inside the endosomal membrane but could inhibit SLO-mediated membrane damage inside the xenophagic membrane (Fig. 3i).

To determine whether the ESCRT machinery also targets the xenophagosomes/xenophagolysosomes of other bacteria, we observed the localization of LC3 and CHMP4B during *S.* Typhimurium and *S. aureus* infection. CHMP4B-EGFP and mCherry-LC3 each surrounded intracellular *Salmonella*, but these signals rarely colocalized (Supplementary Fig. 2a, b), indicating that ESCRT targets *Salmonella*-containing vacuoles (SCV) but not *Salmonella*-containing xenophagic vacuoles. These results were consistent with a previous report that the ESCRT machinery selectively repairs the SCV[41], and suggest that ESCRT is not always necessary for the regulatory process of xenophagolysosomes. On the other hand, CHMP4B-EFGP colocalized with *S. aureus*-containing LC3 vacuoles (Supplementary Fig. 2a, b), suggesting ESCRT-III is recruited to xenophagic vacuoles of cytolysin-secreting bacteria.

### ESCRT machinery maintains the integrity of autophagolysosomes and defends cells against GAS

We next investigated which ESCRT components are required for the maintenance of xenophagolysosomes and xenophagic degradation of bacteria. Knockdown of components ESCRT-I, -II, and -III did not alter the number of GcAV-positive cells, but significantly suppressed the acidification of GcAVs (Supplementary Fig. 3a–c). The kinetics of GcAV acidification showed that ESCRT-I subunit TSG101 is critical for the maintenance of acidic GcAVs after 3 hpi (Supplementary Fig. 3d). Furthermore, knockdown of components ESCRT-I, -II, and -III significantly increased the number of surviving bacteria (Supplementary Fig. 3e, f). These findings suggested that the ESCRT machinery is critical for the integrity of xenophagolysosomes and xenophagic defense against GAS infection.

### Rab41 binds to VPS4 and regulates the recruitment of VPS4 to GcAVs

Given that knockdown or knockout of Rab41 caused the accumulation of ESCRT-III on GcAVs, Rab41 is presumed to be required to dissociate ESCRT-III, but no Rab proteins involved in ESCRT disassembly have been reported. To gain insight into the mechanism through which Rab41 regulates ESCRT-III dissociation, we performed a Co-immunoprecipitation (IP) experiment between Rab41 and ESCRT components during GAS infection. Of the 25 ESCRT proteins, the ESCRT-I subunits (VPS28, VPS37A, VPS37D, MVB12A, MVB12B, UBAP1), the ESCRT-II subunits (EAP20, EAP30), and VPS4 were precipitated with Rab41, indicating that Rab41 interacts with ESCRT proteins (Fig. 4a). We then examined the involvement of Rab41 in the recruitment of ESCRT-I and VPS4 to GcAVs. We found that recruitment of VPS28 (ESCRT-I) to GcAV was not suppressed (Fig. 4b, c), but that of

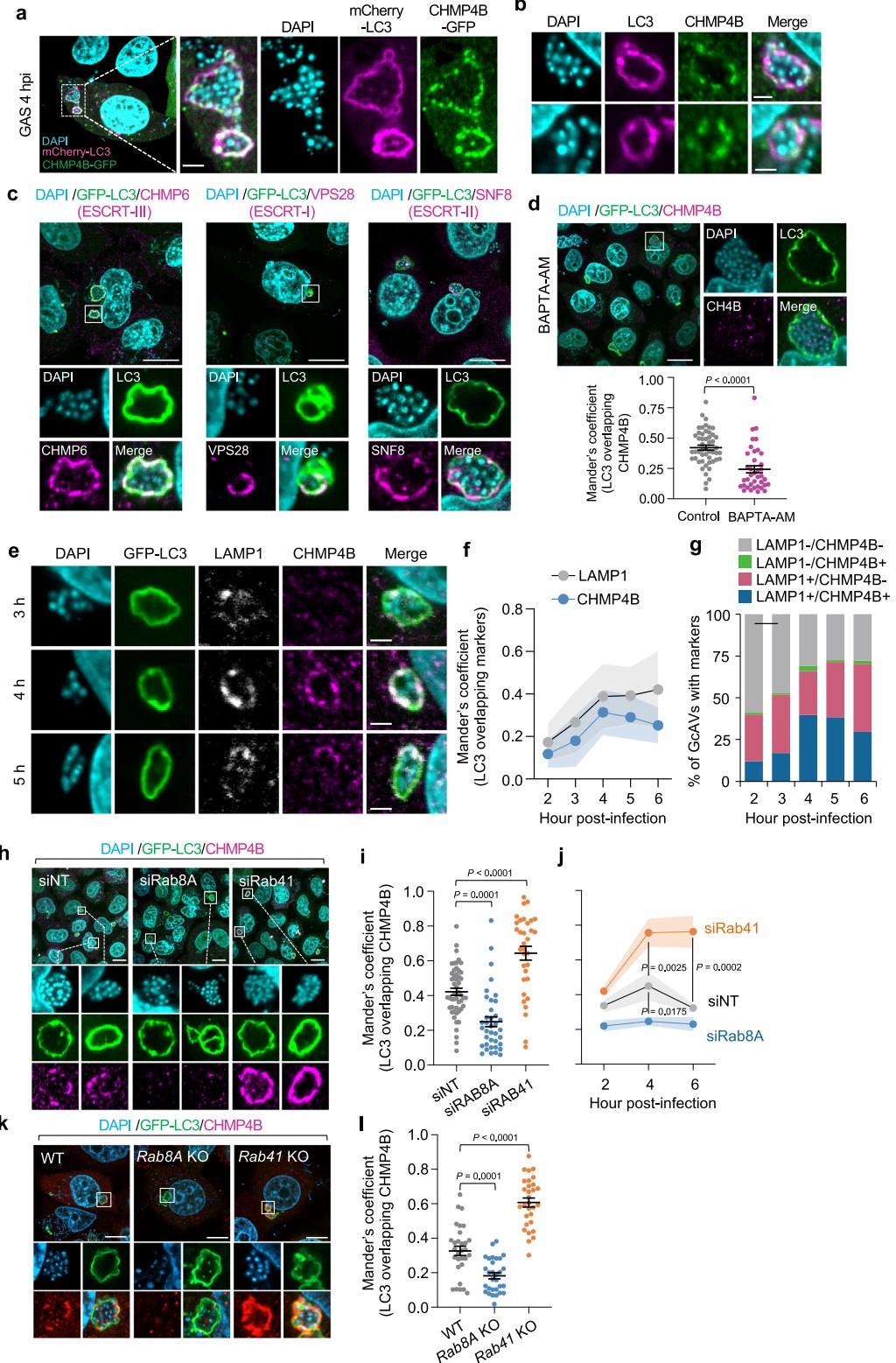

VPS4 was significantly decreased in *Rab41*-knockout cells compared with wild-type cells (Fig. 4d, e). In addition, we found that exogenic expression of the wild-type of Rab41 and its GTPase mutants in *Rab41*-knockout cells significantly recovered VPS4 recruitment to GcAVs and decreased bacterial survival (Fig. 4d–f). Because VPS4 disassembles ESCRT-III[42], it was suggested that Rab41 contributes to host defense by regulating VPS4 dynamics and, thereby, ESCRT disassembly via a unique GTPase-independent mechanism. To check whether Rab41

GTPase mutants used in this study affect its GTPase-dependent function, we performed RUSH (retention using stacked hock) assay to examine the ER-to-the Golgi trafficking during overexpressing Rab41 mutants. Overexpression of Rab41 T44N (GDP-locked form) significantly decreased ER-to-Golgi transport of the reporter (Supplementary Fig. 4a, b), which is consistent with a previous report.

ESCRT-III proteins assemble helical filament structures that expose their membrane interaction sites on the outside of the filament,

**Fig. 2 | ESCRT targets xenophagolysosomes, and localization of the ESCRT complex is regulated by Rab8A and Rab41. a** Localization of the ESCRT-III component CHMP4B during GAS infection. HeLa cells transfected with CHMP4B-EGFP and mCherry-LC3 were infected with GAS for 4 h, fixed, then stained with DAPI. Shown are representative confocal single slice images of three independent experiments. Scale bar, 2 µm. **b** HeLa cells were infected with GAS for 4 h, fixed, then immunostained for endogenous LC3 and CHMP4B. Cellular and bacterial DNA were stained with DAPI. Shown confocal images are single slice and representative of three independent experiments. Scale bar, 2 µm. **c** HeLa cells stably expressing GFP-LC3 were infected with GAS for 4 h, fixed, then immunostained for endogenous CHMP6, VPS28, or SNF8. Cellular and bacterial DNA were stained with DAPI. Shown confocal images are single slice and representative of three independent experiments. Scale bar, 10 µm. **d** HeLa cells stably expressing GFP-LC3 were infected with GAS in absence or presence of BAPTA-AM (10 µM) for 4 h, fixed, then immunostained for endogenous CHMP4B. Cellular and bacterial DNA were stained with DAPI. Representative confocal single slice images and quantification of CHMP4B recruitment to GcAVs. Scale bar, 10 µm. The proportion of the LC3 signal (GcAVs) overlapping the CHMP4B signal were quantified by Mander's coefficient M1. Data are individual values and mean ± SEM ($n = 50$ GcAVs examined over three independent experiments). Unpaired two-tailed $t$ test. **e–g** HeLa cells stably expressing GFP-LC3 were infected with GAS for the indicated time, fixed, then immunostained for endogenous LAMP1 and CHMP4B. Cellular and bacterial DNA were stained with DAPI. Representative confocal micrographs of single slice (**e**) and the proportion of the LC3 signal (GcAVs) overlapping the LAMP1 or CHMP4B signal were quantified by Mander's coefficient M1 (**f**). Scale bar, 2 µm. Data are mean (circles) ± SEM (shadows) ($n = 20$ GcAVs examined in each three independent experiments). The percentage of GcAV-positive cells for LAMP1 and CHMP4B were counted using microscopy (**g**). **h–j** HeLa cells stably expressing GFP-LC3 were transfected with the indicated siRNA, and infected with GAS. Cells were fixed, immunostained for endogenous CHMP4B and stained with DAPI. Representative confocal micrographs of single slice (**h**), and the proportion of the LC3 signal (GcAVs) overlapping CHMP4B was quantified by Mander's coefficient M1 (**i, j**). Scale bar, 10 µm. Data in (**i, j**) are individual values and the mean ± SEM (30 GcAVs examined over three independent experiments). One-way ANOVA, Dunnett's test. **k, l** HeLa WT, *Rab8A* KO, *Rab41* KO cells stably expressing GFP-LC3 were infected with GAS for 4 h, fixed, and immunostained for endogenous CHMP4B. Representative confocal images of single slice (**k**) and the proportion of LC3 signal (GcAVs) overlapping CHMP4B quantified by Mander's coefficient M1 (**l**). Scale bar, 10 µm. Data in (**l**) are individual values and the mean ± SEM (30 GcAVs examined over three independent experiments). One-way ANOVA, Dunnett's test. Source data are provided as a Source Data file.

and the AAA-ATPase VPS4 binds on the inside of the filament and disassembles the ESCRT-III filaments upon ATP hydrolysis[43–45]. Therefore, it was surprising that the Rab protein regulates VPS4 recruitment. To check whether ESCRT-0, -I, -II, and -III are also required for VPS4 recruitment to GcAVs, we knocked down the expression of ESCRT components and quantified VPS4 signals (immunostained for VPS4A/B) on GcAVs. Knockdown of PDCD6IP, TSG101, CHMP2A, and CHMP3 increased the VPS4 signal on GcAVs. Still, none of the ESCRT knockdown mutants suppressed the recruitment of VPS4 onto GcAVs, suggesting that ESCRT-I, -II, and -III are not critical for the recruitment of VPS4 (Supplementary Fig. 5a, b).

Our immunoprecipitation analyses demonstrated an endogenous interaction between Rab41 and VPS4 in non-infected and infected conditions (Fig. 4g). Mammals harbor two paralogues, VPS4A and VPS4B. Rab41 interacted with both VPS4A and VPS4B (Supplementary Fig. 6a). To understand the molecular mechanism by which Rab41 regulates VPS4 recruitment, we mapped the interaction region of VPS4A with Rab41. VPS4A contains microtubule interacting and trafficking (MIT), large AAA (LA), α7, small AAA (SA), α9, and C-terminal region (CTR) (Fig. 4h). We next examined the interaction of Rab41 with these domain deletion mutants of VPS4A and found that the LA domain is critical for the interaction with Rab41 (Fig. 4i). We further checked the interaction of each domain construct of VPS4A with Rab41. The LA domain preferentially precipitated with Rab41, indicating that Rab41 interacts with VPS4A via its LA domain (Fig. 4j). To confirm the recruitment of VPS4 through Rab41, we observed the EGFP tagged VPS4A FL (full-length), LA domain, and MIT domain during GAS infection. VPS4 interacts with ESCRT-III components via the MIT domain[46], but the MIT domain did not show GcAV localization, and the LA domain, which binds to Rab41, clearly colocalized with GcAVs (Fig. 4k), supporting the theory that VPS4 is recruited to xenophagolysosomes through attaching to Rab41. We next examined if VPS4 associates with Rab41 in an ATPase activity-dependent manner, but the inactive form of VPS4A (VPS4A E228Q) also interacted with Rab41 to the same extent as wild-type VPS4A (Supplementary Fig. 6b). We next examined whether Rab41 directly binds to VPS4A by a pull-down assay and found that recombinant Rab41 directly binds to VPS4A (Supplementary Fig. 6c). To determine whether the Rab41–VPS4 interaction depends on the nucleotide-bound status of Rab41, we used GDP-locked T44N and GTP-locked Q89L mutants of Rab41; however, both mutants of Rab41 precipitated VPS4 (Supplementary Fig. 6c). Phosphorylation of Rab GTPase is known

as a possible regulatory mechanism and the phosphorylation sites are located in conserved Rab family/Rab subfamily motif and complementarity determining regions. We next substituted candidate phosphorylation site S94 to A (Alanine) in Rab41 and found that S94A mutation reduced the interaction with endogenous VPS4 (Fig. 4l). Furthermore, S94 residue was critical for Rab41-mediated inhibition of bacterial survival (Fig. 4m), suggesting that Rab41 exerts its bactericidal activity through interaction with VPS4, possibly via phosphorylation of Rab41. These findings demonstrated that Rab41 directly binds to VPS4 in a GTPase-independent manner, and Rab41 regulates the recruitment of VPS4 to GcAVs.

## Rab41 targets damaged autophagic membranes

Rab41 is a Rab6-like protein reported to be localized to the cytosol and Golgi apparatus, where it is thought to regulate Golgi organization and endoplasmic reticulum (ER)–Golgi trafficking[47,48]; however, its localization and function during bacterial infection remains undefined. We expressed EmGFP-tagged Rab41 and infected cells with GAS. EmGFP-Rab41 showed cytosolic localization in non-infected cells and accumulated to GcAVs after 4 hpi (Fig. 5a). Furthermore, the kinetics revealed that only less than 5% of GcAVs were Rab41-positive at 2 hpi, but Rab41-positive GcAVs at 5 hpi or later reached 40% (Fig. 5b). Because ESCRT-III is recruited to GcAVs from 2 to 4 hpi (Fig. 5b), it was suggested that the targeting of Rab41 to GcAVs occurred later than ESCRT-III recruitment. To observe the localization of endogenously expressed Rab41, we generated an EmGFP-Rab41 knockin HeLa cell line by CRISPR/Cas9 genome editing (Supplementary Fig. 7a, b). We found that endogenous EmGFP-Rab41 (eEmGFP-Rab41) colocalized with CHMP4B around bacteria (Fig. 5c). We next investigated whether the recruitment of Rab41 to GcAVs was GTPase-dependent and found that both the GDP-locked and GTP-locked mutants of Rab41 were recruited to GcAVs (Fig. 5d, e). We also checked Rab41 localization in xenophagy-deficient *ATG7*-knockout cells and Rab41-positive bacteria were not observed in *ATG7*-knockout cells at 1−5 hpi (Fig. 5f). Taken together, these data suggested that Rab41 targets xenophagolysosomes via a GTPase-independent mechanism.

To check whether Rab41 is redistributed to damaged endomembranes in response to membrane injury, we observed Rab41 localization when cells were treated with the lysosomal damaging agent Leu-Leu-O-Me (LLOMe), which is converted to its membranolytic form through cathepsin D within lysosomes. As expected, EmGFP-Rab41 was redistributed to lysosomes in response to LLOMe treatment and

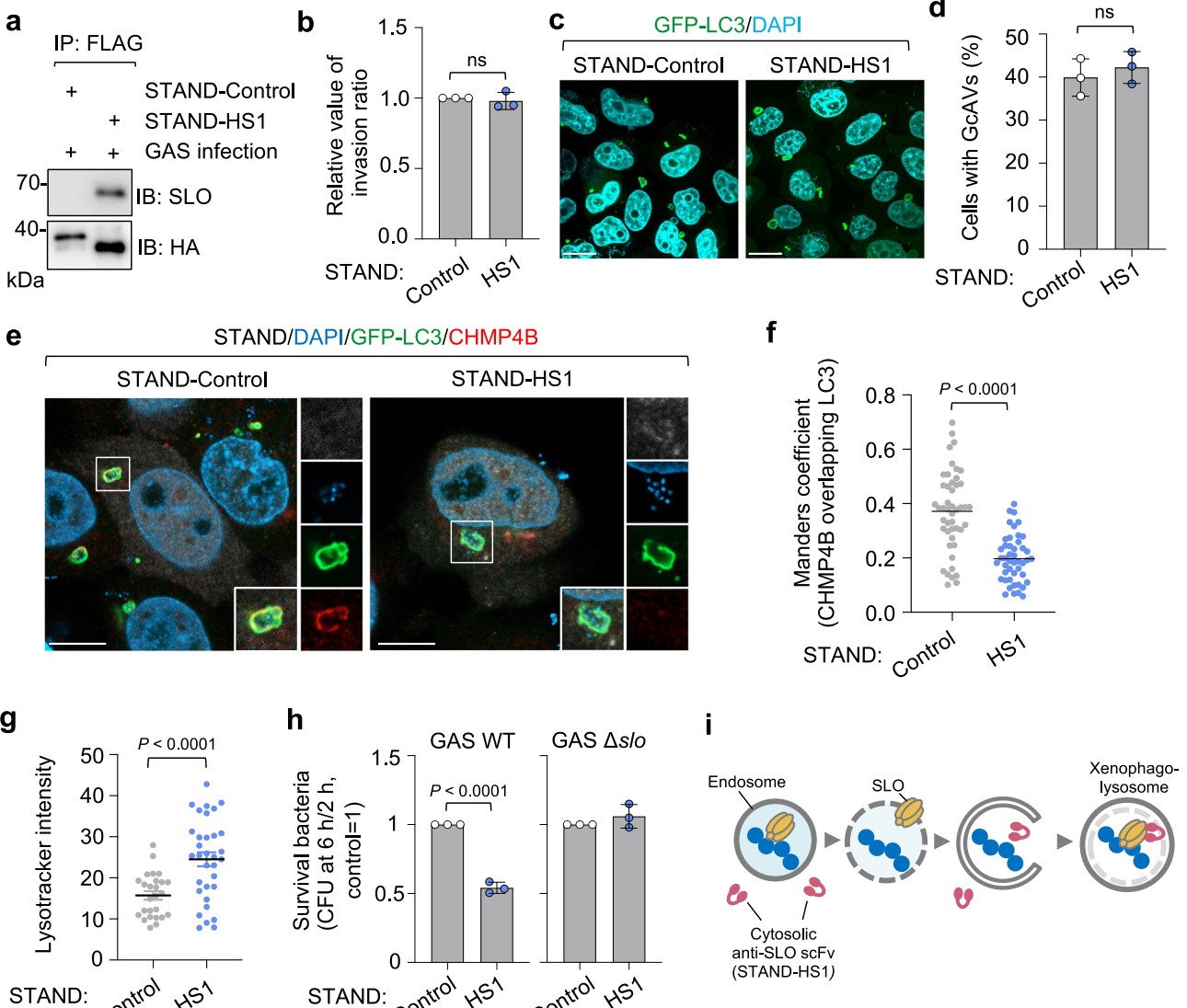

**Fig. 3 | GAS secretion cytolysin SLO triggers ESCRT recruitment to xenophagolysosomes and promotes intracellular proliferation. a** HeLa cells expressing STAND-HyHEL10 (Control) or STAND-HS1 were infected with GAS for 4 h, and STANDs were precipitated with anti-FLAG antibody and Protein G Sepharose. SLO and STANDs were detected from the precipitated samples by western blotting. Show data are representative of three independent experiments. **b** HeLa cells expressing STAND-control or STAND-HS1 were infected with GAS and invasion efficiencies were determined by colony counting. Data are individual values and mean ± SEM (*n* = 3 biologically independent experiments). Unpaired two-tailed *t* test. ns not significant (0.05 < *P*). **c, d** HeLa cells expressing GFP-LC3 and either STAND-control or STAND-HS1 were infected with GAS for 4 h, fixed, and stained with DAPI. Representative confocal images of single slice are shown (**c**) and cells with GcAVs were quantified (**d**). Scale bar, 10 μm. Shown data are individual values and mean ± SEM (*n* = 3 biologically independent experiments). Unpaired two-tailed *t* test. ns not significant (0.05 < *P*). **e, f** The recruitment of CHMP4B to GcAVs in STAND-expressing cells. HeLa cells expressing GFP-LC3 and either STAND-control or STAND-HS1 were infected with GAS for 4 h, fixed, and immunostained for endogenous CHMP4B. Bacterial and cellular DNA were stained with DAPI. Representative confocal images of single slice (**e**) and quantification of CHMP4B

recruitment to GcAVs (**f**). Scale bar, 10 μm. The proportion of LC3 (GcAVs) overlapping CHMP4B was quantified by Mander's coefficient M1. Data in (**f**) are individual values and the mean (50 GcAVs <*n* examined over three independent experiments). Unpaired two-tailed *t* test. **g** The acidification of GcAVs in STAND-expressing cells. Quantification of the LysoTracker intensity in GcAVs. HeLa cells expressing STAND plasmids were infected with GAS for 4 h and fixed. Cells were stained with LysoTracker Red 30 min prior to fixation. The LysoTracker intensity inside GcAVs was measured. Data are individual values and the mean ± SEM (30 GcAVs <*n* examined over three independent experiments). Unpaired two-tailed *t* test. **h** HeLa cells transfected with the indicated STANDs were infected with GAS JRS4 wild-type (WT) or isogenic *slo* deletion mutants (Δ*slo*) and the intracellular bacterial CFU was determined at 6 hpi. Data are individual values and mean ± SEM (*n* = 3 biologically independent experiments). Unpaired two-tailed *t* test. **i** Proposed model that STAND-HA inhibits xenophagolysosomal damages. STAND-HS1 is expressed in cytosol, therefore they do not access to SLO secreted inside endosomes. STAND-HS1 can be incorporated into xenophagic vacuoles and inhibits SLO-mediated damages of xenophagolysosomes. Source data are provided as a Source Data file.

colocalized with CHMP4B (Fig. 5g–i), suggesting that Rab41 targets damaged endomembrane. We further checked the GTPase dependence of Rab41 localization on damaged lysosomes. Similar to the results of recruitment to GcAV, Rab41 localized to damaged lysosomes in a GTPase-independent manner (Fig. 5j). We also show that eEmGFP-

Rab41 redistributed to lysosomal compartment upon LLOMe treatment (Fig. 5k, l). A time-course analysis of endogenous CHMP4B, LC3, LGALS3, Rab41, and LAMP1 localization after lysosomal damage showed that CHMP4B-positive lysosomes were visible as early as 10 min after LLOMe stimulation and gradually disappeared after

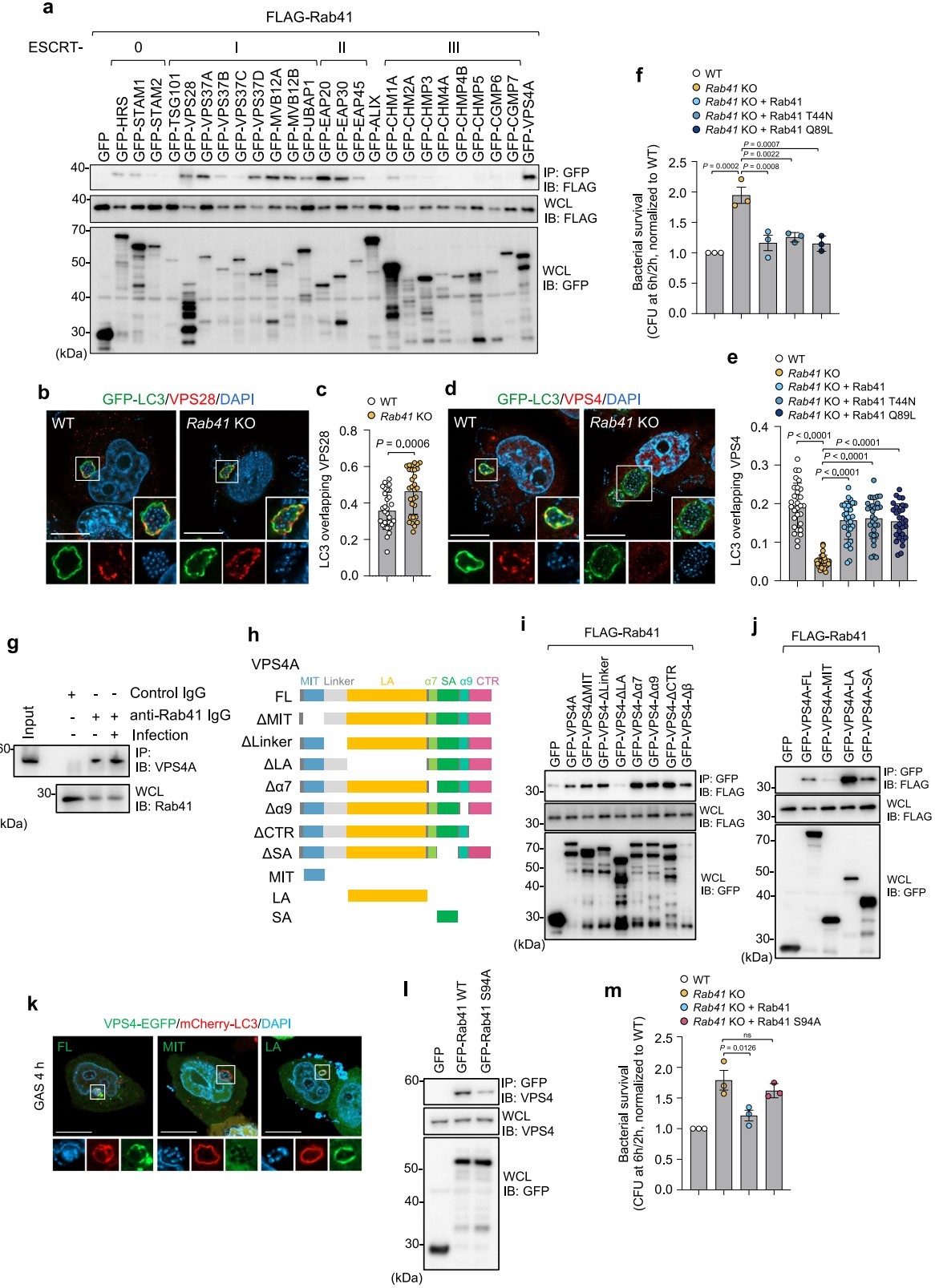

20 min, LC3 and LGALS3 were recruited from 10 to 20 min, whereas Rab41 recruitment to lysosomes occurred after 20 min, which is after LC3 and LGALS3 recruitment (Fig. 5m). Since the recruitment of ESCRT machinery occurs before that of LGALS3 and the lysophagy machinery, these results suggested that Rab41 is recruited to injured membrane later than lysophagy machinery.

## Rab 41 is critical for the homeostasis of autolysosomes

To determine whether Rab41 is involved in ESCRT-mediated repair and lysosomal homeostasis, we depleted Rab41 and observed CHMP4B dynamics. CHMP4B-positive small vacuoles appeared 10 min after LLOMe treatment and decreased after 60 min in control cells (Fig. 6a, b). By contrast, VPS4A-depleted cells harbored numerous

**Fig. 4 | Rab41 binds to VPS4 and is required for VPS4 localization to xenophagolysosomes. a** Co-immunoprecipitation between FLAG-Rab41 and GFP-ESCRT proteins during GAS infection. HeLa cells expressing GFP tagged ESCRT components and FLAG-Rab41 were infected with GAS for 4 h. Cell extract were immunoprecipitated (IP) with GFP-Trap beads. The resulting samples, IP and whole cell lysate (WCL), were analyzed by immunoblot analysis. Data shown are representative of three independent experiments. **b, c** The recruitment of ESCRT-I subunit VPS28 to GcAVs in *Rab41*-knockout cells. HeLa WT and *Rab41* knockout (KO) cells were infected with GAS for 4 h, fixed, and immunostained for endogenous VPS28. Cellular and bacterial DNA were stained with DAPI. The proportion of the LC3 vacuole signal (GcAVs) overlapping VPS28 was quantified by Mander's coefficient M1. Representative confocal images of single slice (**b**) and quantification of VPS4 recruitment to GcAVs (**c**). Data are individual values and mean ± SEM ($n = 30$ GcAVs examined over three independent experiments). Unpaired two-tailed *t* test. Scale bar, 10 μm. **d, e** The recruitment of VPS4 to GcAVs in *Rab41*-knockout cells. HeLa WT and *Rab41* knockout cells expressing FLAG-Rab41, -Rab41 T44N, or -Q89L were infected with GAS for 4 h, fixed, and immunostained for endogenous VPS4. Cellular and bacterial DNA were stained with DAPI. The proportion of the LC3 vacuole signal (GcAVs) overlapping VPS4 was quantified by Mander's coefficient M1. Representative confocal images of single slice (**d**) and quantification of VPS4 recruitment to GcAVs (**e**). Scale bar, 10 μm. Data are individual values and mean ± SEM ($n = 30$ GcAVs examined over three independent experiments). One-way ANOVA, Tukey's test. **f** HeLa WT and *Rab41* knockout cells expressing FLAG-Rab41, -Rab41 T44N, or -Q89L were infected with GAS JRS4 and the intracellular bacterial CFU was determined at 6 hpi. Data are individual values and mean ± SEM ($n = 3$ biologically independent experiments). One-way ANOVA, Tukey's test. **g** Immunoprecipitation analysis. HeLa cells were infected with GAS for 4 h and then lysed with IP lysis buffer. Cell lysate were immunoprecipitated with anti-Rab41 or control rabbit antibody. The resulting samples were analyzed by immunoblot analysis. Data shown are representative of three independent experiments. **h** Domain organization and deletion mutants of VPS4. **i, j** Co-immunoprecipitation analysis. HEK293T were transfected with GFP tagged VPS4 deletion mutants and FLAG tagged Rab41, and then cell lysates were immunoprecipitated with GFP-Trap beads. The resulting samples were analyzed by immunoblot analysis. Data shown are representative of three independent experiments. **k** HeLa cells transfected with VPS4-EGFP FL, MIT or LA domain and mCherry-LC3 were infected with GAS for 4 h, fixed and stained with DAPI. Shown are representative confocal single-slice images of three independent experiments. Scale bar, 10 μm. **l** Co-immunoprecipitation analysis. HEK293T were transfected with GFP tagged Rab41 wild-type or Rab41 S94A mutant, and then cell lysates were immunoprecipitated with GFP-Trap beads. The resulting samples were analyzed by immunoblot analysis. Data shown are representative of three independent experiments. **m** HeLa WT and Rab41 knockout cells expressing GFP-Rab41 constructs were infected with GAS, and intracellular survival bacteria were determined by colony counting. Data are individual values and mean ± SEM ($n = 3$ biologically independent experiments). One-way ANOVA, Tukey's test. Source data are provided as a Source Data file.

small CHMP4B dots even after 60 min and showed large CHMP4B-positive vacuoles (Fig. 6a–c). Notably, in Rab41 knockdown cells, small CHMP4B puncta reduced from 10 to 60 min after LLOMe treatment, but enlarged CHMP4B-positive vacuoles were accumulated (Fig. 6a–c). These enlarged vacuoles were positive for both LC3 and LAMP1 (Fig. 6d), suggesting that the knockdown of Rab41 caused enlargement of autolysosomes.

To confirm the involvement of Rab41 in the maintenance of autolysosomes, we observed the localization of LC3, CHMP4B, and VPS4 in *Rab41*-knockout cells. Knockout of *Rab41* also exhibited CHMP4B-positive large LC3 vacuoles after 60 min LLOMe-treatment (Fig. 6e). VPS4 partially colocalized with LC3 small puncta, but enlarged LC3 vacuoles lacked VPS4 signal in *Rab41*-knockout cells (Fig. 6e). Taken together, it was suggested that Rab41 is dispensable for the early phase of lysosomal repair, but required for later autolysosomal homeostasis through VPS4 (Fig. 6f). We next confirmed whether the defect of ATPase activity in VPS4 also leads to enlarged autolysosome formation by the overexpression of VPS4 E228Q. Cells with VPS4A E228Q showed enlarged LAMP1-positive vacuoles after LLOMe treatment, indicating that VPS4 ATPase activity is required for autolysosomal homeostasis (Supplementary Fig. 8a). Conversely, overexpression of GDP-locked or GTP-locked mutants of Rab41 did not induce enlarged CHMP4B-positive lysosomes in LLOMe-treated cells (Supplementary Fig. 8b). Taken together, these findings suggested that Rab41 plays an essential role in autolysosomal homeostasis, similar to VPS4 ATPase, through its GTPase activity-independent function.

## Rab41 interacts with TOM1L2 on the damaged lysosomal membrane

We next investigated how Rab41 selectively targets damaged autolysosomes and xenophagolysosomes via a GTPase-independent mechanism. Rab GTPases are localized to distinct membrane compartments and tightly bound to the membrane due to double prenylation of the C-terminus[49]. However, Rab41 does not possess a C-terminal prenylation motif[48], suggesting that Rab41 does not anchor to the membrane via prenylation. Indeed, Rab41 that lacks the C-terminal region (CTR) localizes to autolysosomes upon LLOMe stimulation (Supplementary Fig. 9a, b). Rab41 targets damaged auto-/xenophagolysosomes independently of its nucleotide-bound form and lipidation. We then sought to identify the protein that recruits Rab41 to damaged membranes. We performed IP-mass spectrometry (IP-MS)

analysis of GFP-Rab41 protein complexes in cells stimulated with LLOMe (Fig. 7a). HEK293T cells expressing GFP or GFP-Rab41 were treated with LLOMe and GFP or GFP-Rab41 was precipitated with GFP-Trap beads, and precipitated proteins were identified by mass spectrometry ($n = 4$). The proteins identified in the GFP-only condition or showing the highest abundance in GFP-only condition were removed from subsequent data analysis. We found that seven proteins (TOM1L2, KCTD19, CLTA, CLTC, RNF5, RNA185, and MYH9) were significantly enriched as binding proteins of Rab41 upon LLOMe stimulation (Fig. 7a). VPS4A was also detected in both LLOMe non-treated and treated conditions (Supplementary Data 1, Identified protein number: IP00697). TOM1L2 is an ancestral ESCRT-0 protein and is suggested to be involved in ciliary trafficking[50]. EmGFP-TOM1L2 showed cytosolic localization in control cells and redistributed to mCherry-LGALS3-positive vacuoles (Fig. 7b). In addition, EmGFP-TOM1L2 and endogenous TOM1L2 colocalized with GcAVs during GAS infection (Fig. 7c).

We checked the endogenous interaction between Rab41 and TOM1L2, and Co-IP experiments validated the Rab41–TOM1L2 interaction only in LLOMe-treated cells (Fig. 7d). To examine the subcellular localization of the Rab41–TOM1L2 complex, we visualized their interaction sites using a proximity ligation assay (PLA). PLA signals were colocalized with GcAVs, suggesting Rab41 binds to TOM1L2 on xenophagolysosomes (Fig. 7e).

TOM1L2 contains VHS (VPS27-Hrs-STAM) and GAT (GGA and TOM1) domains, which are thought to mediate functions such as adaptor protein and ubiquitin-binding protein, respectively[51]. To examine whether these domains are required for targeting the damaged autolysosomes, we observed the recruitment of TOM1L2 domain deletion mutants in GAS-infected or LLOMe-treated cells. Deletion of VHS or GAT completely abolished the recruitment of TOM1L2 to damaged lysosomes and GcAVs (Fig. 7f), indicating that TOM1L2 targets damaged membranes through VHS and GAT domains.

## TOM1L2 recruits Rab41 to xenophagolysosomes

We next examined whether TOM1L2 is involved in targeting Rab41 to GcAVs, and found that knockdown of TOM1L2 expression significantly decreased Rab41-positive GcAVs (Fig. 8a, b and Supplementary Fig. 10a). In addition, the recruitment of VPS4 to GcAVs was suppressed by TOM1L2 depletion (Fig. 8c, d). To confirm the defect in membrane repair by TOM1L2 depletion, we monitored the acidification of GcAVs. The LysoTracker intensity inside GcAVs was decreased

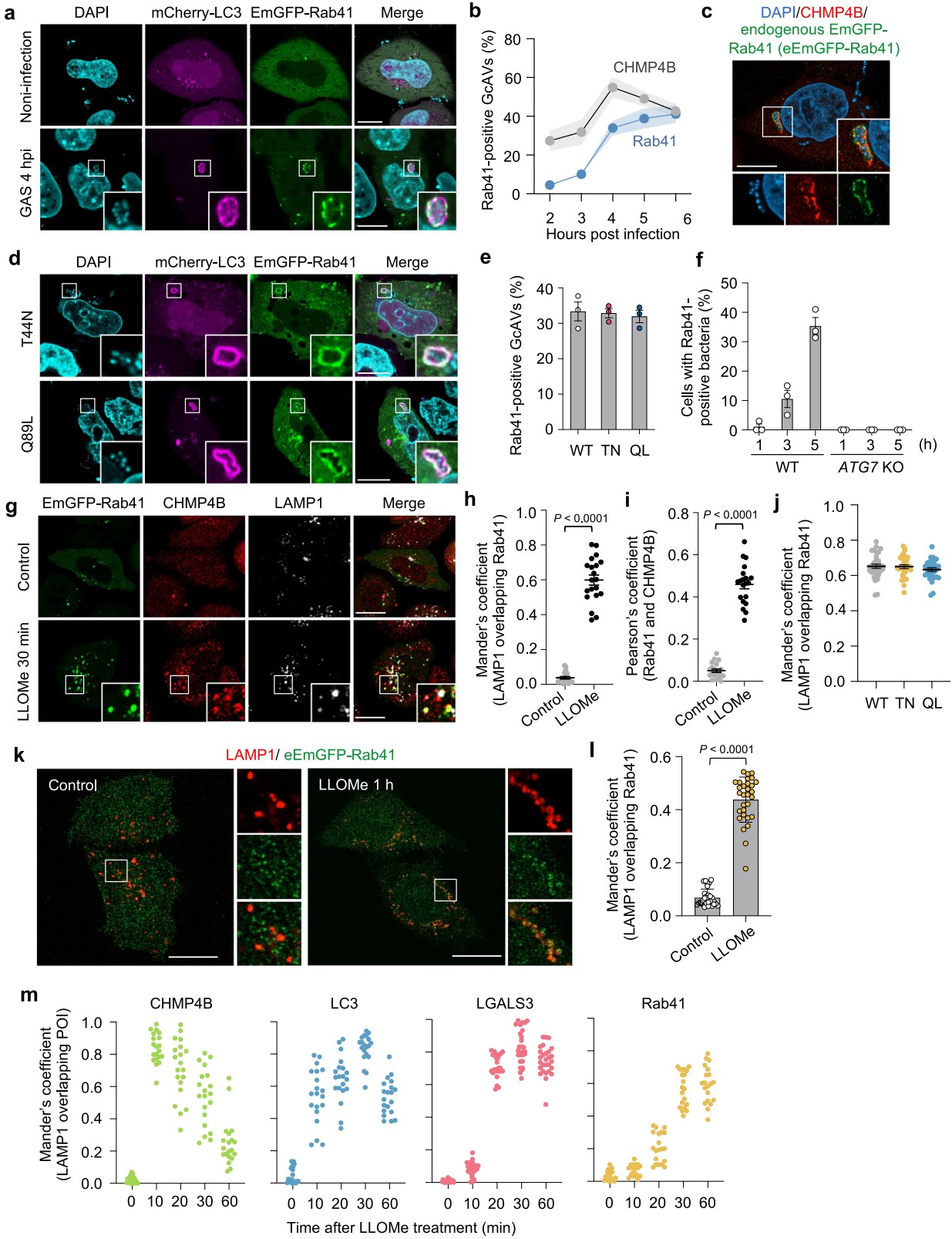

after 3 hpi (Fig. 8e, f). To further validate this, we examined bacterial survival, and the number of viable intracellular GAS was increased in TOM1L2-knockdown cells (Fig. 8g). Similar results were obtained in *TOM1L2*-knockout cells (Fig. 8h, i–k and Supplementary Fig. 10b). Together, these findings showed that TOM1L2 is necessary for Rab41-mediated xenophagy homeostasis.

## Discussion

In the present study, we revealed that Rab GTPases, such as Rab8A and Rab41, are involved in the repair of xenophagolysosomes through the ESCRT machinery (Fig. 8l). Furthermore, we identified TOM1L2 as an adaptor protein between damaged xenophagolysosomes or auto-lysosomes and the Rab41–VPS4 complex. Depletion of Rab8A, Rab41,

**Fig. 5 | Rab41 targets damaged xenophagolysosomes and lysosomes, and is critical for lysosomal homeostasis. a, b** Rab41 recruitment to GcAVs. HeLa cells expressing EmGFP-Rab41 and mCherry-LC3 were infected with GAS for 4 h, fixed, and stained with DAPI. (**a**) Representative confocal single slice images of EmGFP-Rab41 localization in uninfected or GAS-infected cells and (**b**) the time course of Rab41 and CHMP4B recruitment to GcAVs. Data in (**b**) represent are mean (circles) ± SEM (shadow) from three independent experiments and 50 GcAVs <n were counted in each condition. Scale bars, 10 μm. **c** HeLa GFP-knockin cells (endogenous EmGFP-Rab41, eEmGFP-Rab41) were infected with GAS for 4 h, fixed, and immunostained for endogenous CHMP4B. Cellular and bacterial DNA were stained with DAPI. Shown are representative confocal single-slice images and representative of three independent experiments. Scale bar, 10 μm. **d, e** The recruitment of Rab41 GTPase mutants to GcAVs. HeLa cells expressing EmGFP-Rab41 mutants and mCherry-LC3 were infected with GAS for 4 h, fixed, and stained with DAPI. Representative confocal single slice images (**d**) and quantification of Rab41-positive GcAVs (**e**). Scale bar, 10 μm. Data in (**e**) are individual values and mean ± SEM from three independent experiments and 50 GcAVs <n were counted in each condition. **f** HeLa ATG7-knockout cells expressing EmGFP-Rab41 and mCherry-LC3 were infected with GAS for 1, 3, or 5 h, fixed and stained with DAPI. Shown are quantification of EmGFP-Rab41-positive GcAVs. Data are individual values and mean ± SEM from three independent experiments and 200 cells <n were counted in each condition. **g–i** HeLa cells expressing EmGFP-Rab41 were incubated with 1 mM LLOMe for 30 min, fixed, and immunostained for endogenous CHMP4B and LAMP1.

Representative confocal single-slice images (**g**), quantification of Rab41 recruitment to lysosomes (**h**), and quantification of the colocalization between Rab41 and CHMP4B (**i**). Scale bar, 10 μm. Data in (**h**) are the proportion of LAMP1 signal overlapping Rab41 were quantified using Mander's coefficient M1. Individual values and mean ± SEM are shown (n = 30 GcAVs examined over three independent experiments). Data in (**i**) are colocalization efficiency of Rab41 and CHMP4B were quantified using Pearson's coefficient. Individual values and mean ± SEM are shown (n = 30 GcAVs examined over three independent experiments). Unpaired two-tailed t test. **j** HeLa cells expressing EmGFP-Rab41 T44N or Q89L were treated with LLOMe for 30 min, fixed and immunostained for endogenous LAMP1. Data are the proportion of LAMP1 signal overlapping Rab41 were quantified using Mander's coefficient M1. Individual values and mean ± SEM are shown (n = 30 GcAVs examined over three independent experiments). **k, l** HeLa EmGFP knockin (eEmGFP-Rab41) cells were treated with 1 mM of LLOMe, and immunostained for endogenous LAMP1. Representative confocal single-slice images (**k**) and quantification of the recruitment of endogenous Rab41 to lysosomes (**l**). Scale bar, 10 μm. Data in (**l**) represent individual values and mean ± SEM (n = 30 GcAVs examined over three independent experiments). Unpaired two-tailed t test. **m** HeLa wild-type cells or eEmGFP-Rab41 knockin cells were incubated with 1 mM LLOMe, fixed at different time points as indicated, and immunostained for endogenous LAMP1 and CHMP4B, LC3, or LGALS3. Shown data are individual values determined by Mander's coefficient M1 (n = 20 cells examined over 3 independent experiments). Source data are provided as a Source Data file.

and ESCRT proteins increased bacterial survival, indicating that the maintenance of xenophagolysosomes facilitates bacterial degradation. By contrast, inhibition of bacterial cytolysin SLO using an intrabody, enhanced GcAV acidification and suppressed the intracellular proliferation of bacteria. These results indicated that some intracellular GAS could evade xenophagy by reducing acidification within xenophagolysosomes. Of note, Rab41 is the first Rab protein identified that directly interacts with the ESCRT machinery, but its function does not seem to be dependent on GTPase activity, indicating that Rab proteins, the master regulators of membrane transport, may have GTPase-independent parts.

In our siRNA screen in this study, the knockdown of 10 Rab proteins increased the number of surviving intracellular bacteria. However, this did not rule out the possible involvement of other Rab proteins in the bactericidal pathway. We showed that the knockdown of Rab19 decreased xenophagosome formation and its lysosomal fusion. Rab19 has been suggested to interact with the ATG16 complex and be involved in xenophagy of *S. pneumoniae*[31,52]. Thus, Rab19 might be a common regulator in xenophagy. We also showed that the large Rab GTPase Rab44 is required for xenophagosome–lysosome fusion and bacterial degradation. Although the molecular mechanisms underlying Rab44-mediated lysosomal fusion remain unknown, because Rab44 regulates the microtubule-dependent retrograde pathway[53,54], Rab44 is likely required to transport xenophagosomes to lysosomes. Besides Rab19, several other Rab proteins, such as Rab33B and Rab37, have been reported to regulate ATG proteins directly[55–57], but these do not appear to be required for xenophagy against GAS. Given the complexity of membrane regulation in xenophagy, it is likely that there are other unidentified ATG-regulating Rab proteins. Further analysis is required to understand the molecular mechanisms of membrane regulation in xenophagy.

Our study focused on xenophagic vacuoles and showed that xenophagolysosomes are maintained through the ESCRT machinery. Rab8A-mediated ESCRT machinery is reported to repair the phagosomal membrane damaged by bacterial pathogens in macrophages[10]. However, both exogenous and endogenous EmGFP-Rab41 targeted only LC3-positive bacteria-containing vacuoles, and knockout of ATG7 abolished Rab41 recruitment to GAS. Therefore, it is likely that Rab41 functions in the homeostasis of autophagic lysosomal vacuoles. Why is Rab41 not involved in recruiting VPS4 to lysosomes and its repair, but only needed for xenophagolysosomes repair? The endosomal membrane damage induces xenophagy (lysophagy), which can be

eliminated, but for the membrane damage of the last-resort xenophagolysosome, another immune system should be activated to protect the cells. Of note, RNF5 and RNF185 were identified to interact with Rab41 upon lysosomal membrane damage (Fig. 7a). Because RNF5 and RNF185 have been reported to facilitate various innate immune responses to pathogens[58–60], it would be informative to reveal how Rab41 associates with these E3 ligases and their contribution to host defense systems.

Here, we concluded that the Rab8A- and Rab41-mediated ESCRT system repairs xenophagolysosomes based on two main findings: (i) knockdown or knockout of Rab8A, Rab41, or ESCRT proteins caused a reduction in the acidification inside GcAVs, and (ii) inhibition of SLO by an intrabody decreased the recruitment of ESCRT to GcAVs and enhanced acidification of GcAVs. However, the ESCRT machinery has been reported to be involved in various steps of autophagy. Recent studies have shown that ESCRT components are transiently recruited to nascent autophagosomes with residence times of a few minutes and regulate the sealing process to form closed autophagosomes during starvation-induced autophagy and mitophagy[61–63]. Therefore, there is a possibility that the ESCRT machinery is required for the closing step of xenophagosomes, but because ESCRT proteins did not localize transiently at one site of xenophagosomes but instead surrounded the entire membrane for several hours in response to cytolysin, our conclusion that ESCRT functions to repair the xenophagolysosomal membrane appears reasonable. In addition, consistent with previous reports, the depletion of ESCRT proteins caused the accumulation of LC3-positive structures, but these structures were not observed in Rab8A- or Rab41-knockdown cells, suggesting that Rab8A and Rab41 are not involved in other functions of ESCRT in autophagy.

We found that Rab41 directly binds to VPS4 and is required to recruit VPS4 to xenophagolysosomes. VPS4 is an AAA-ATPase that is critical for the disassembly of the ESCRT-III filament[42]. VPS4 interacts with CHMP3 and CHMP4 proteins through its MIT domain, and its localization is also regulated by ESCRT proteins[13]. Therefore, it was unexpected that VPS4 recruitment involves Rab41. Because the recruitment of VPS4 to xenophagolysosomes was not affected by the knockdown of either ESCRT protein and not MIT but LA domain was enough to target to xenophagolysosomes, the localization route of VPS4 appears to be distinct from that of ESCRT-I, -II, and -III in response to autolysosomal damage. We showed that Rab41 targets

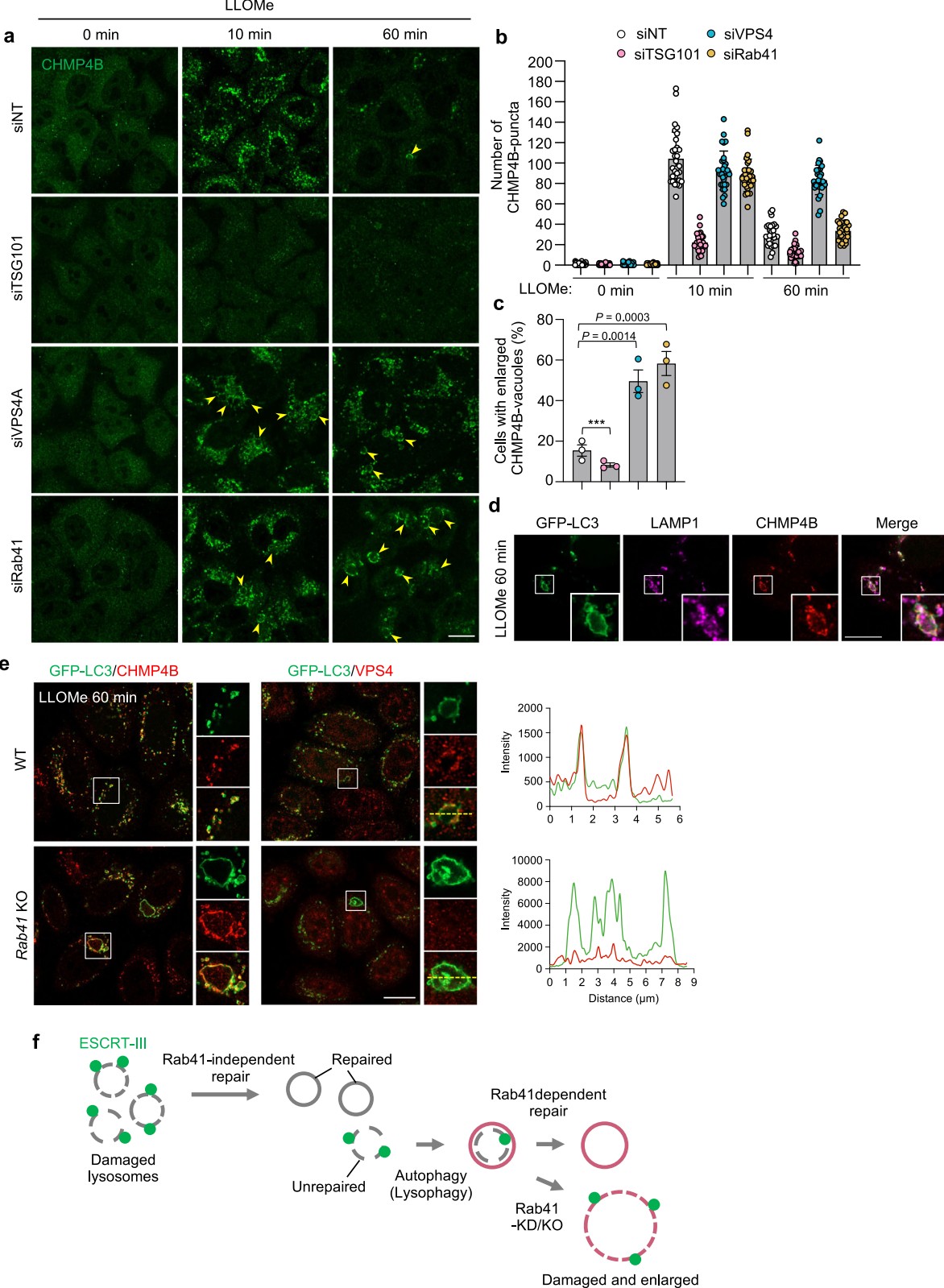

autolysosomes and interacts with VPS4 in a GTPase-independent manner. Rab41 is reportedly involved in Golgi organization and ER–Golgi trafficking via a GTPase-dependent mechanism[47,48], Rab41 appears to have functions other than a canonical molecular switch in membrane trafficking. Some GTPase-independent functions have been described previously. For example, the Rho GTPase Rac1 acts independently of its GTPase activity as an adaptor protein for mTOR and

regulates cellular membrane associations of both mTORC1 and mTORC2 complexes[64]. Rab32 and Rab21 have been implicated in binding to mTOR kinase and integrin receptors, respectively, regardless of the GTP/GDP nucleotide status[65,66]. However, the GTP-bound form of these Rabs was required for their membrane localization and function. Given that Rab41 lacks a prenylation motif, which is critical for anchoring to the membrane, Rab41 may be involved in membrane

**Fig. 6 | Rab41 is required for autolysosomal integrity through VPS4. a–c** HeLa cells transfected with indicated siRNA were treated with LLOMe and fixed at indicated time points, and immunostained for endogenous CHMP4B. Representative confocal single-slice images are shown (**a**). Enlarged CHMP4B-positive vacuoles are indicated by yellow arrows. Scale bar, 10 μm. The numbers of CHMP4B puncta were quantified from randomly-selected cells using imageJ, and data are individual values and mean ± SEM (*n* = 50 GcAVs examined over three independent experiments). One-way ANOVA, Tukey's test. **d** HeLa cells stably expressing GFP-LC3 were treated with LLOMe for 1 h, fixed, and then immunostained for LAMP1 and

CHMP4B. Shown are representative confocal single-slice images of three independent experiments. Scale bar, 10 μm. **e** HeLa WT or *Rab41* knockout cells stably expressing GFP-LC3 were treated with LLOMe for 1 h, fixed, and then immunostained for CHMP4B or VPS4. Shown are representative confocal single-slice images of three independent experiments. Fluorescent line scans across the yellow dashed line in the merged insets. Scale bar, 10 μm. **f** Schematic depicting Rab41-independent lysosomal repair and -dependent autolysosomal repair. Source data are provided as a Source Data file.

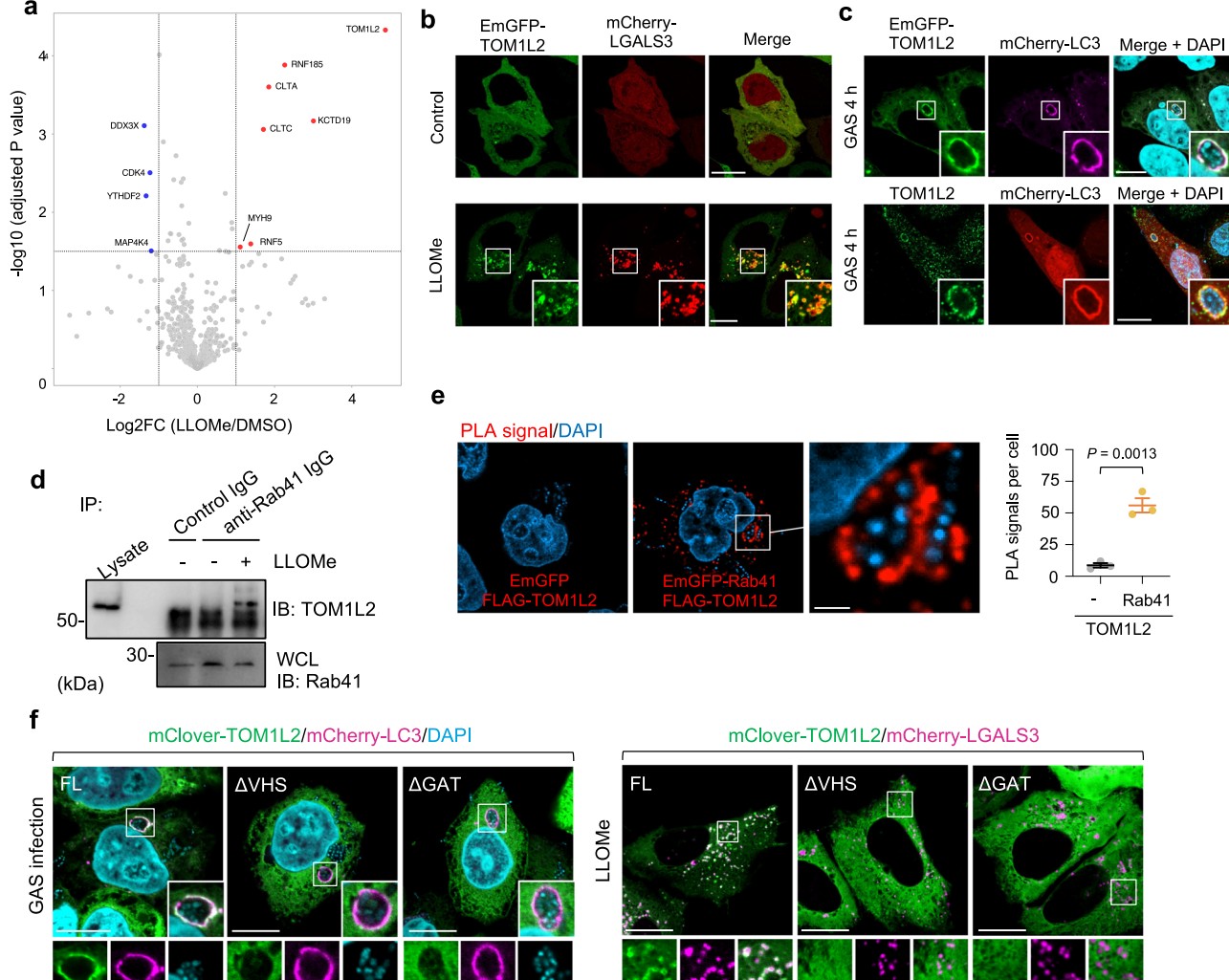

**Fig. 7 | TOM1L2 is localized to damaged autolysosomes and interacts with Rab41. a** Volcano plot showing the subset of proteins with enriched in GFP-Rab41 precipitates. The red and blue plots show proteins that are twofold higher or lower abundance in the LLOMe treatment relative to the DMSO treatment, respectively, with the significance of enrichment (−log10 of adjusted *P* value) (*n* = 4 biological replicates). HEK293T cells expressing GFP or GFP-Rab41 were treated with LLOMe and GFP or GFP-Rab41 was precipitated with GFP-Trap beads, and precipitated proteins were identified by mass spectrometry. Unpaired two-tailed *t* test. **b** HeLa cells expressing EmGFP-TOM1L2 and mCherry-LGALS3 were treated with 1 mM of LLOMe for 30 min. Shown are representative confocal single-slice images of three independent experiments. Scale bar, 10 μm. **c** HeLa cells expressing EmGFP-TOM1L2 and mCherry-LC3 or expressing EmGFP-LC3 were infected with GAS for 4 h, and endogenous TOM1L2 was stained with an anti-TOM1L2 antibody. Shown are representative confocal single-slice images of three independent experiments.

Cellular and bacterial DNA were stained with DAPI. Scale bar, 10 μm. **d** Co-immunoprecipitation assay. HeLa cells were incubated in the presence or absence of LLOMe for 1 h and cell lysates were immunoprecipitated with anti-Rab41 antibody. The resulting samples were analyzed by immunoblot. Data shown are representative of three independent experiments. **e** HeLa cells expressing EmGFP-Rab41 and FLAG-TOM1L2 were infected with GAS for 4 h, and immunostained for GFP and FLAG to assess Rab41-TOM1L2 binding by Duolink PLA. Representative confocal single-slice images and quantification of the PLA signals. Scale bar, 2 μm. Data are individual values and mean ± SEM (*n* = 50 GcAVs examined over three independent experiments). Unpaired two-tailed *t* test. **f** Localization of TOM1L2 domain deletion constructs. HeLa cells were transfected with mCherry-LC3 and mClover-TOM1L2 full-length (FL), ΔVHS, or ΔGAT. Shown are representative confocal single-slice images of three independent experiments. Scale bar, 10 μm. Source data are provided as a Source Data file.

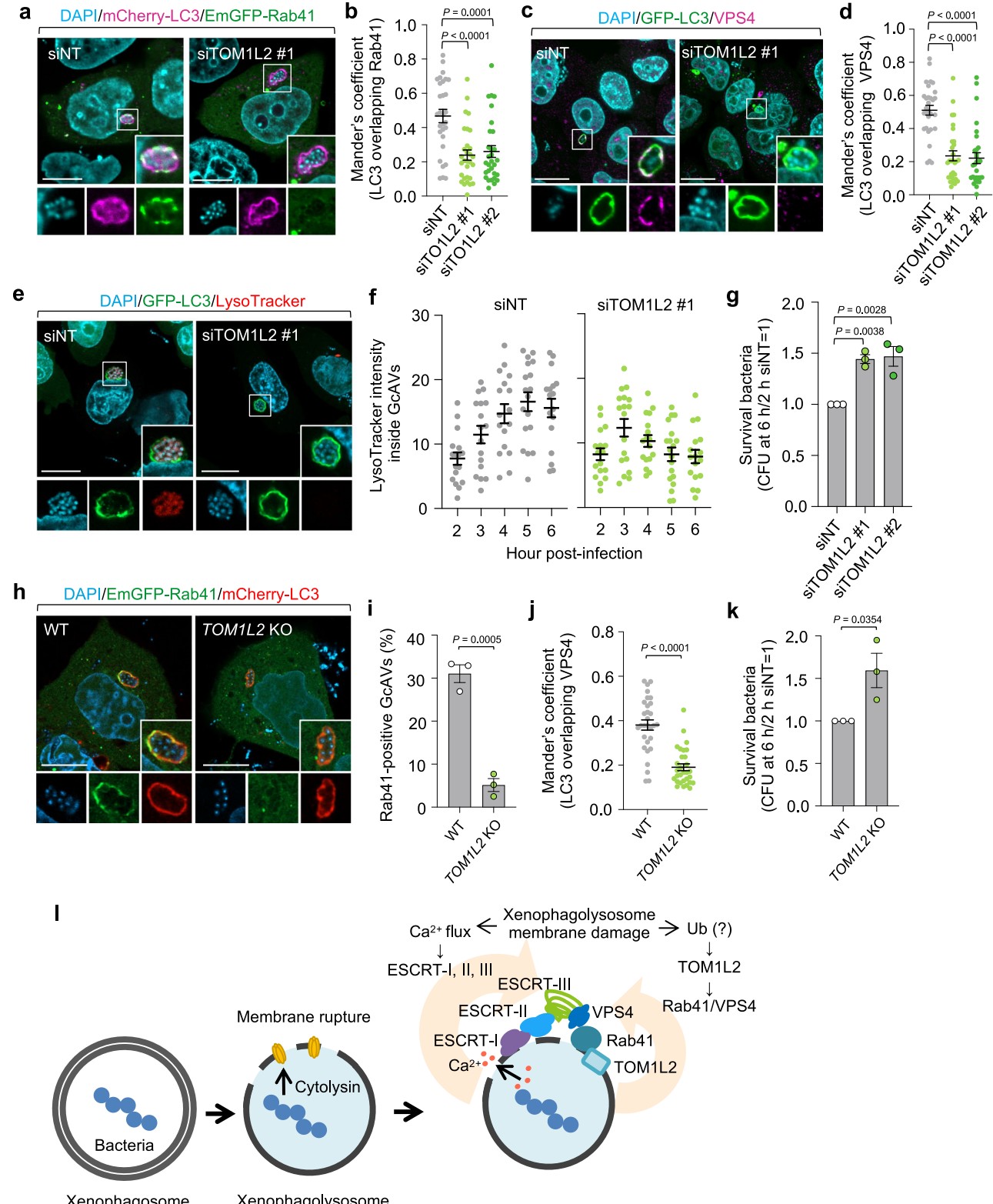

regulation through a mechanism that is mainly different from other Rab proteins. We also identified S94 residue in Rab41 as a critical amino acid for interacting with VPS4 and host defense against GAS infection. S94 in Rab41 is located in Rab family 3 motif and S or T in this conserved motif are frequently phosphorylated by kinases such as LRRK2 and TBK1[67–69]. As far as we examined, the interaction between Rab41 and VPS4 was not infection stimulus-dependent, but investigating whether the interaction is mediated S94 phosphorylation and the

identification of the responsible kinase would be essential to understand the defense mechanism against bacterial infection.

We also identified TOM1L2 as an adaptor protein of Rab41. TOM1L2 is a VHS domain protein family member, but its function is less defined[51]. Other VHS domain proteins, such as HRS and STAM1/2, mediate the sorting of proteins for lysosomal degradation. These proteins act primarily at the sorting endosome and are considered ESCRT-0 proteins. In addition, TOM1 and TOM1L1 also have been

**Fig. 8 | TOM1L2 recruits Rab41 to damaged xenophagolysosomes and is required for xenophagy of GAS. a, b** HeLa cells expressing EmGFP-Rab41 and mCherry-LC3 were transfected with the indicated siRNA, and infected with GAS for 4 h. Cellular and bacterial DNA were stained with DAPI. Representative confocal single-slice images (**a**) and quantification of Rab41 recruitment to GcAVs (**b**). Scale bar, 10 µm. Data in (**b**) are individual values and mean ± SEM ($n = 30$ GcAVs examined over three independent experiments). One-way ANOVA, Dunnett's test. **c, d** HeLa cells expressing GFP-LC3 were transfected with the indicated siRNA, and infected with GAS for 4 h. Endogenous VPS4 was visualized with an anti-VPS4 antibody, and cellular and bacterial DNA were stained with DAPI. Representative confocal single-slice images (**c**) and quantification of VPS4 recruitment to GcAVs (**d**). Scale bar, 10 µm. Data in (**b**) are individual values and mean ± SEM ($n = 30$ GcAVs examined over three independent experiments). One-way ANOVA, Dunnett's test. **e, f** HeLa cells expressing GFP-LC3 were transfected with the indicated siRNA, and infected with GAS. Cells were stained with LysoTracker Red 30 min prior to fixation. Representative confocal single-slice images are presented (**e**) and the intensity of LysoTracker within GcAVs was quantified. Scale bar, 10 µm. Data in (**f**) are individual values and mean ± SEM ($n = 30$ GcAVs examined over three

independent experiments). **g** Intracellular bacterial CFU at 6 hpi in the control and TOM1L2-knockdown cells. Data are individual values and mean ± SEM ($n = 3$ biologically independent experiments). One-way ANOVA, Dunnett's test. **h, i** HeLa WT or *TOM1L2* knockout cells expressing mCherry-LC3 and EmGFP-Rab41 were infected with GAS for 4 h, fixed, and stained with DAPI. Representative confocal single-slice images (**h**) and quantification of Rab41-positive GcAVs (**i**). Scale bar, 10 µm. Data in (**i**) are individual values and mean ± SEM from three independent experiments, and 200 cells <n were counted in each condition. Unpaired two-tailed t test. **j** HeLa WT or *TOM1L2* knockout cells expressing GFP-LC3 were infected with GAS for 4 h. Endogenous VPS4 was visualized with an anti-VPS4 antibody, and cellular and bacterial DNA were stained with DAPI. Shown data are quantification of VPS4 recruitment to GcAVs. Data are individual values and mean ± SEM ($n = 30$ GcAVs examined over three independent experiments). Unpaired two-tailed t test. **k** Intracellular bacterial CFU at 6 hpi in WT and *TOM1L2*-KO cells. Data are individual values and mean ± SEM ($n = 3$ biologically independent experiments). Unpaired two-tailed t test. Source data are provided as a Source Data file. **l** Model of xenophagosome repair system through ESCRT and Rab41-mediated VPS4 dynamics.

---

suggested to be involved in the ESCRT system. TOM1 and TOM1L2 are about 59% identical, but because TOM1 does not localize to xenophagolysosomes[70], the function of ESCRT-mediated repair in xenophagy may be limited to TOM1L2. TOM1 has been shown to interact with ubiquitin through its GAT domain, recruit clathrin onto the endosome, and interact with ubiquitinated proteins, resulting in lysosomal targeting[71,72]. In our proteome analysis of Rab41-interacting protein, clathrin was detected as Rab41-interacting candidate when LLOMe was treated. In addition, E3 ligases RNF5 and RNF185 were identified upon LLOMe treatment (Fig. 7a), and therefore these proteins might be involved in TOM1L1 and Rab41 dynamics. A report shows that *Tom1l2* hypomorphic mice exhibit increased infectious incidence, especially skin infections[73]. Considering the suggested functions of TOM1L2 in this study, TOM1L2 seems to be an essential adaptor protein in response to endomembrane damage by bacterial pathogens, such as GAS and *S. aureus*.

We demonstrated that not only xenophagolysosomes but also LLOMe-induced autolysosomes are targeted by Rab41, and Rab41 is required for the homeostasis of autolysosomes. However, it is not clear whether this function serves any physiological functions other than immunity. This study only showed that Rab41 contributes to autolysosome homeostasis after prolonged treatment with high concentrations of LLOMe. Therefore, further detailed studies will be needed to conclude the involvement of Rab41 in physiological function through VPS4.

In conclusion, our data highlighted the novel interplay between bacterial pathogen and host defense machinery and identified Rab41 and TOM1L2 as regulators of VPS4. Our knowledge of Rab41 and TOM1L2 remains limited, but the functional significance of these proteins has been demonstrated in this study. Future detailed functional analysis of these proteins will advance our understanding of host defense mechanisms.

## Methods
### Cell culture and transfection
HeLa cells were obtained from American Type Culture Collection and maintained in 5% $CO_2$ at 37 °C in Dulbecco's modified Eagle's medium (Nacalai Tesque) supplemented with 10% fetal bovine serum (Gibco) and 50 µg/mL gentamicin (Nacalai Tesque). HeLa cells tested negative for mycoplasma using the MycoStrip (Invivogen, Cat #: 20596-84).

### Drugs
LysoTracker Red DND-99 (Thermo Fisher Scientific), BAPTA-AM (Dojindo), and L-Leucyl-L-Leucine methyl ester (Cayman Chemical) were purchased.

### Plasmids and siRNA transfection
Human HRS, STAM1, STAM2, TSG101, VPS28, VPS37A, VPS37B, VPS37C, VPS37D, MVB12A, MVB12B, UBAP1, EAP20, EAP30, EAP45, PDCP6IP, VPS4A, VPS4B, TOM1L1, and TOM1L2 were amplified by polymerase chain reaction (PCR) from total mRNA derived from HEK293T cells were introduced into pcDNA-6.2/N-EmGFP-DEST, pcDNA-6.2/N-3xFLAG-DEST, pcDNA-6.2/N-mCherry-DEST, pGEX-6P-1-DEST, pcDNA-6.2/N-mClover-DEST, and pLenti6/V5-DEST using Gateway technology (Invitrogen). Human CHMP1A, CHMP2A, CHMP3, CHMP4A, CHMP4B, CHMP5, CHMP6, and CHMP7 were amplified by PCR from total mRNA derived from HeLa cells and were introduced into pEGFP-N1 (Clontech). Human Rab41 were amplified by PCR from total mRNA derived from HeLa cells and were introduced into pcDNA-6.2/N-3xFLAG-DEST, pcDNA-6.2/N-EmGFP-DEST using Gateway technology (Invitrogen). Rab41, VPS4A, and TOM1L2 were mutated by site-directed mutagenesis using a PrimeSTAR mutagenesis basal kit (Takara Bio). In STAND-HyHEL10 (LC369677) and HS1 (LC545573.1; IgH and LC545574.1; IgK) construction, the variable regions of immunoglobulin heavy- and light-chain were fused by a glycine/serine (GGGGS)3 linker and synthesized by FASMAC. Synthesized VH-linker-VL were amplified by PCR to conjugate with human influenza hemagglutinin (HA) tag at C-terminal and introduced to pcDNA-6.2/N-3xFLAG-DEST. For knockdown experiments, HeLa cells were seeded at $4 \times 10^4$ cell/well in 24-well plate and 10 pmol siRNA oligonucleotides Rab1A (s229381; Thermo Fisher Scientific), Rab1B (s119; Thermo Fisher Scientific), Rab2A (s11660; Thermo Fisher Scientific), Rab2B (s39689; Thermo Fisher Scientific), Rab3A (s11668; Thermo Fisher Scientific), Rab3B (s11671; Thermo Fisher Scientific), Rab3C (s41884; Thermo Fisher Scientific), Rab3D (s18327; Thermo Fisher Scientific), Rab4A (s11675; Thermo Fisher Scientific), Rab4B (s28802; Thermo Fisher Scientific), Rab5A (s11679; Thermo Fisher Scientific), Rab5B (s11681; Thermo Fisher Scientific), Rab5C (s11710; Thermo Fisher Scientific), Rab6A (s11685; Thermo Fisher Scientific), Rab6B (s28326; Thermo Fisher Scientific), Rab7A (s15444; Thermo Fisher Scientific), Rab7B (s50335; Thermo Fisher Scientific), Rab8A (s8681; Thermo Fisher Scientific), Rab8B (s28634; Thermo Fisher Scientific), Rab9A (s17916; Thermo Fisher Scientific), Rab9B (s27693; Thermo Fisher Scientific), Rab10 (s21391; Thermo Fisher Scientific), Rab11A (s16703; Thermo Fisher Scientific), Rab11B (s17649; Thermo Fisher Scientific), Rab12 (s47369; Thermo Fisher Scientific), Rab13 (s11692; Thermo Fisher Scientific), Rab14 (s28311; Thermo Fisher Scientific), Rab15 (s51763; Thermo Fisher Scientific), Rab17 (s34609; Thermo Fisher Scientific), Rab18 (s22705; Thermo Fisher Scientific), Rab19 (s53589; Thermo Fisher Scientific), Rab20 (s31158; Thermo Fisher Scientific), Rab21 (s22823; Thermo Fisher Scientific), Rab22A (s32994; Thermo Fisher

Scientific), Rab23 (s28568; Thermo Fisher Scientific), Rab24 (s28804; Thermo Fisher Scientific), Rab25 (s32702; Thermo Fisher Scientific), Rab26 (s24590; Thermo Fisher Scientific), Rab27A (s11693; Thermo Fisher Scientific), Rab27B (s11696; Thermo Fisher Scientific), Rab28 (s17912; Thermo Fisher Scientific), Rab29 (s17083; Thermo Fisher Scientific), Rab30 (s26138; Thermo Fisher Scientific), Rab31 (s21731; Thermo Fisher Scientific), Rab32 (s21619; Thermo Fisher Scientific), Rab33A (s17909; Thermo Fisher Scientific), Rab33B (s37939; Thermo Fisher Scientific), Rab34 (s38246; Thermo Fisher Scientific), Rab35 (s21708; Thermo Fisher Scientific), Rab36 (s18462; Thermo Fisher Scientific), Rab37 (s223876; Thermo Fisher Scientific), Rab38 (s24321; Thermo Fisher Scientific), Rab39A (s42009; Thermo Fisher Scientific), Rab39B (s44481; Thermo Fisher Scientific), Rab40A (s21588; Thermo Fisher Scientific), Rab40B (s21588; Thermo Fisher Scientific), Rab40C (s33730; Thermo Fisher Scientific), Rab41 (s51269; Thermo Fisher Scientific), Rab42 (s41783; Thermo Fisher Scientific), Rab43 (s50453; Thermo Fisher Scientific), Rab44 (s53538; Thermo Fisher Scientific), RASEF (s46039; Thermo Fisher Scientific), PDCD6IP (s19465; Thermo Fisher Scientific), TSG101 (s14439; Thermo Fisher Scientific), VPS28 (s27577; Thermo Fisher Scientific), VPS37A (s44037; Thermo Fisher Scientific), VPS25 (s38895; Thermo Fisher Scientific), SNF8 (s22247; Thermo Fisher Scientific), VPS36 (s27276; Thermo Fisher Scientific), CHMP1A (s533441; Thermo Fisher Scientific), CHMP2A (s26027;Thermo Fisher Scientific), CHMP3 (s28474; Thermo Fisher Scientific), CHMP4A (s26441; Thermo Fisher Scientific), CHMP5 (s28237; Thermo Fisher Scientific), CHMP6 (s35990; Thermo Fisher Scientific), CHMP7 (s40780; Thermo Fisher Scientific), VPS4A (s25966;Thermo Fisher Scientific), VPS4B (s18273; Thermo Fisher Scientific) or non-targeting siRNAs (4390843, 12935300, Thermo Fisher Scientific) was transfected using Lipofectamine 3000 (Invitrogen) 24 h later. Forty-eight hours after transfection, cells were used for experiments.

## Bacterial infection
GAS strain JRS4 (M6$^+$F1$^+$) was grown in Todd–Hewitt broth (BD Diagnostic Systems) supplemented with 0.2% yeast extract. Cell cultures in media without antibiotics were infected for 1 h at a multiplicity of infection of 100 for confocal microscopic analysis and 10 for the bacterial viability assay. Infected cells were washed with phosphate-buffered saline (PBS) and treated with 100 μg/mL gentamicin for an appropriate time to kill bacteria that were not internalized.

## Bacterial viability assay
HeLa cells were cultured in 24-well culture plates and infected as described in the section "Bacterial infection". After an appropriate incubation period, cells were lysed in sterile distilled water, and then serial dilutions of the lysates were plated on THY agar plates. Colony counting was used to determine the numbers of invading and surviving GAS; the bacterial survival data are presented as the ratio of "intracellular live GAS at 6 h" to "total intracellular GAS at 2 h."

## Fluorescence microscopy
For immunofluorescence, the following antibodies were used: LAMP1 (H4A3; sc-20011; Santa Cruz Biotechnology, 1:100), CHMP4B (13683; Proteintech Group, 1:200), CHMP6 (16278-1-AP; Proteintech Group, 1:100), VPS28 (15478-1-AP; Proteintech Group, 1:100), SNF8 (67696-1-IG; Proteintech Group, 1:100), VPS4A/B (17673-1-AP, Proteintech Group, 1:100), LC3 (ab51520, Abcam, 1:100), LGALS3 (556904, BD Pharmingen, 1:100), and TOM1L2 (GTX106295; GENETEX, 1:100). Anti-mouse or anti-rabbit IgG conjugated to AlexaFluor 488 and 594 (Molecular Probes, Eugene, OR, USA), and anti-mouse or rabbit IgG conjugated to AlexaFluor 647 (Jackson Laboratories) were used as a secondary antibody for immunostaining. Cells were washed with PBS, fixed for 15 min with 4% PFA in PBS, permeabilized with 0.1% Triton in PBS for 10 min, washed with PBS, and blocked at room temperature for

1 h with skim milk (5% skim milk, 2.5% goat serum, 2.5% donkey serum, and 0.1% gelatin in PBS) or BSA (2% BSA and 0.02% sodium azide in PBS). Cells were then probed at room temperature for 1 h with the primary antibody diluted in blocking solution, washed with PBS, and labeled with the secondary antibody. To visualize bacterial and cellular DNA, cells were stained with 4',6-diamidino-2-phenylindole (DAPI; Dojindo). Confocal fluorescence micrographs were single slice acquired with an LSM900 laser-scanning microscope with a Plan-Apochromat 63 × /1.4 oil DIC objective lens and ZEN software (Zeiss) or FV1000 laser-scanning microscope with an UPlanSApo 100× oil/1.40 objective lens and Fluoview software (Olympus).

## Immunoblot analysis
Cell lysate samples were mixed with equal volume of 2× Laemmli SDS sample buffer and boiled for 5 min. Samples were separated by SDS-PAGE and then transferred to polyvinylidene difluoride (PVDF) membrane. Membrane was blocked with blocking buffer (5% skim milk or Blocking One (Nacalai Tesque)), and incubated with primary antibody diluted with blocking buffer at 4 °C overnight. For immunoblot analysis, the following antibodies were used as a primary antibody: streptolysin O (SLO) (ab188539; Abcam, 1:1000), HA (HA124; Nacalai Tesque, 1:1000), FLAG (M2; A2220; Sigma-Aldrich, 1:1000), GFP (GF200; 04363-24; Nacalai Tesque, 1:1000), MBP (E8032; New England BioLabs, 1:1000), GAPDH (sc-47724; Santa cruz biotechnology), Rab8A (D22D8; 6975 S; Cell Signaling Technology, 1:1000), Rab41 (NBP2-83434; NOVUS Biologicals, 1:1000), Rab41 (18818-1-AP; Proteintech Group, 1:500), VPS4A/B (17673-1-AP, Proteintech Group, 1:1000), and TOM1L2 (GTX106295; GENETEX, 1:1000). After wash with PBS containing 0.1% Tween-20 three times, membrane was incubated with secondary antibody for 2 h at room temperature. Horseradish peroxidase-conjugated anti-rabbit IgG (H + L) (111-005-003; Jackson ImmunoResearch Laboratories), anti-mouse IgG (H + L) (115-005-003; Jackson ImmunoResearch Laboratories), and anti-rabbit IgG (conformation-specific; L27A9; 5127; Cell Signaling Technology) were used as secondary antibodies for immunoblots. After wash three times, the membrane was reacted with Chemi-Lumi One Super (Nacalai Tesque), and images were obtained using LAS-4000 mini Luminescent image analyzer (Fujifilm).

## Immunoprecipitation
Cells were harvested, washed with PBS, and lysed for 20 min on ice in lysis buffer containing 10 mM Tris-HCl (pH 7.4), 150 mM NaCl, 10 mM MgCl$_2$, 1 mM EDTA, 1% Triton X-100, and a proteinase-inhibitor cocktail (Nacalai Tesque). After brief centrifugation, supernatants were reacted for 2 h at 4 °C with GFP-Trap (ChromoTek) or appropriate antibodies, then incubated for another 1 h at 4 °C with shaking. Immunoprecipitates were collected by brief centrifugation, washed five times with lysis buffer, and analyzed by immunoblotting, as described above.

## Protein expression and pull-down assay
GST-tagged proteins constructed in pGEX-6P-1 (GE Healthcare Life Sciences) and MBP-tagged proteins in pMAL-c5x (New England Biolabs) were transformed into *Escherichia coli* BL21 (DE3) cells, which were then cultured at 37 °C in LB medium supplemented with 100 μg/mL ampicillin, and induced for 3 h at 37 °C with 0.3 mM isopropyl β-D-thiogalactopyranoside (Nacalai Tesque). Cells were harvested by centrifugation, washed with PBS, lysed in lysis buffer (40 mM Tris-HCl pH 7.5, 5 mM EDTA, and 0.5% Triton X-100), sonicated, and cleared by centrifugation. The resulting supernatant of GST-tagged protein was incubated with Glutathione Sepharose 4 Fast Flow (GE Healthcare Life Sciences) for 2 h at 4 °C. After several washes with lysis buffer, and beads were mixed and reacted with the supernatant of MBP-tagged proteins for 2 h at 4 °C. After wash five times with lysis buffer, the resulting samples were subjected to SDS-PAGE, and followed by coomassie brilliant blue (CBB) staining or immunoblot.

## Generation of knockout or knockin cell lines by CRISPR/Cas9

CRISPR/Cas9 was used to KO *Rab41* and *TOM1L2*. CRISPR guide (g) RNAs were designed to target an exon common to all splicing variants of the gene of interest (5′-TCTGCCTTTGGTCACGACG-3′ for *Rab41* and 5′- TGAGGATTGGACGTTGAATA-3′ for *TOM1L2*). HeLa cells were transfected with the vector pSpCas9(BB)−2A-Puro (PX459) V2.0 vector (Addgene 62988) containing the CRISPR target sequence. Untransfected cells were removed by selection on plates containing 2 µg/mL puromycin (Invivogen). Single colonies were expanded, and depletion of the target gene was confirmed by immunoblot. As a secondary screen for some KO lines, genomic DNA was isolated, and target regions were amplified by PCR and sequenced to confirm the presence of the desired frameshift insertions and deletions. For generation of N-terminal GFP knock-in HeLa cell lines the Rab41 locus was targeted with an GFP donor DNA (Invitrogen, A42992) with chemically modified single-guide RNA (sgRNA) (CCAGGCCUCGUCGUGACCAA) according to the manufacturer's instruction. Briefly, 240 ng of the sgRNA, 500 ng of the PCR amplified donor DNA, and 1250 ng of Cas9 protein (TrueCut Cas9 V2, Thermo Fisher Scientific) were co-transfected into HeLa cells (5 × 10⁴ cell/well) using Lipofectamine CRISPRMAX reagent (Thermo Fisher Scientific). The following primers were used to amplify donor DNA (fwd: 5′-EFAGCCATTTTGGCTG-CAACCTACCTGAAGCGATGGGAGGTAAGCCCTTGCATTCG-3′; rev 5′-FFAGCCTCCGGCCTCCATCCAGGCCTCGTCGTGACCAAAGGCA-GAACCGCTTCCACTACCTGAACC-3′). The next day, cells were selected in puromycin (0.5 µg/ml) for 5 days, and surviving single colonies were screened by genomic DNA based PCR and Western blot to validate homozygous knock-in of the GFP-tag on the endogenous Rab41 gene.

## Reverse transcribed quantitative PCR

For gene expression analysis, HeLa cells were transfected with siRNA as described above and the total RNA was isolated from the cells using the Quick-RNA Miniprep Kit (Zymo Research). The PrimeScriptII 1st strand cDNA Synthesis Kit (Takara) was used for cDNA synthesis, and specific gene transcripts were quantified using SsoAdvanced Univ SYBR Supermix (Bio-Rad). The primer sets used in this study is listed in Supplementary Data 2. The relative change in transcript levels upon knockdown of the target gene was calculated using the ΔΔCT method with values normalized to GAPDH.

## In situ proximity ligation assay

The proximity ligation assay was performed with Duolink (Olink Bioscience). HeLa cells grown on coverslips were infected with or without GAS for 4 h, fixed in 4% paraformaldehyde (PFA) for 15 min, washed with PBS, permeabilized with 0.1% Triton X-100, and blocked with 2% BSA blocking buffer for 1 h at room temperature. Cells were probed overnight at 4 °C with primary antibodies diluted in 2% BSA blocking buffer. Labeling with secondary antibodies, ligation, and signal amplification was conducted as recommended by the manufacturer. Proximity ligation assay dots were imaged with an LSM900 laser-scanning microscope (Zeiss).

## RUSH assay

To assess anterograde transport, HeLa cells were transfected with the RUSH plasmid (Streptavidin-KDEL_ManII-SBP-mCherry) (Addgene 65253), and at 24 h post-transfection, 40 µM biotin (Nacalai Tesque) was added to the cells, and after incubation for 1 h, the cells were fixed with 4% PFA (15 min) and immunostained with anti-GM130 antibody. Images were acquired using confocal microscopy and analyzed using the ImageJ software. The proportion of mCherry-ManII signal overlapping GM130 was quantified using Mander's coefficient M1; >30 cells were analyzed in each condition.

## IP-mass spectrometry

For the identification of proteins interacting with Rab41, we performed MS experiment of IP samples collected from DMSO- or LLOMe-treatment with replicates (n = 4 for each condition). 25 µL eluates were reduced with Tris (2-carboxyethyl) phosphine, alkylated with methyl methane-thiosulfonate, digested with trypsin and purified using S-Trap micro spin columns (ProtiFi, Farmingdale NY) according to the manufacturer's instruction. Purified protein digests were evaporated and resuspended in 0.1% formic acid. The recovered protein digests were separated using Nano-LC-Ultra 2D-plus (Eksigent) with trap column (200 µm × 0.5 mm ChromXP C18-CL 3 µm 120 Å (Eksigent)) and analytical column (75 µm × 15 cm ChromXP C18-CL 3 µm 120 Å (Eksigent)) in a trap and elute mode. The solvents used for the separation were 0.1% formic acid/water (mobile phase A) and 0.1% formic acid/acetonitrile (mobile phase B). The binary gradient used for the separation was as follows; 2 to 33.2 % B in 125 min, 33.2 to 98% B in 2 min, 98% B for 5 min, 98 to 2% B in 0.1 min, and 2% B for 17.9 min. The flow rate was 300 nL/min. The analytical column was set to 40 °C. The eluates were directly infused to TripleTOF 5600+ System coupled with a NanoSpray III source and heated interface (SCIEX, Framingham, MA, USA) and ionized in an electrospray ionization-positive mode. Data acquisition was carried out with an information-dependent acquisition method. The acquired datasets were analyzed using ProteinPilot software version 5.0.1 (SCIEX) for peptide and protein identification, with Uni-ProtKB/Swiss-Prot database for human (April 2022) appended with known common contaminants database (SCIEX). Reliabilities of peptide identification was evaluated by the confidence scores calculated by ProteinPilot and quality of the database search was confirmed by the false discovery rate analysis in which the reversed amino acid sequences were used as decoy. Relative abundances of the identified proteins were estimated on the platform of Progenesis QI for Proteomics software version 4.2 (Nonlinear Dynamics, Newcastle upon Tyne, UK). Raw data files were imported to generate an aggregate, and the peptide identification results by ProteinPilot, with confidence at least 95%, were used for assignment. Label-free quantification of proteins were performed using Hi-N method employing protein grouping (Nonlinear Dynamics)[74]. The proteins identified in the GFP-only condition were excluded. In addition, among proteins commonly identified in three conditions (GFP-only, DMSO treatment, and LLOMe treatment), proteins showing the highest abundance in GFP-only condition were removed from subsequent data analysis. The identified proteins and the relevant data were listed in Supplementary Data 1.

## Statistical analysis

Cells containing LC3 or other markers were quantified through direct visualization using a confocal microscope (Zeiss or Olympus). Unless otherwise indicated, 200 GAS-infected cells were examined in each experiment, and at least three independent experiments were performed for each trial. Values, including those plotted, represent the mean ± standard error of the mean (SEM). Pearson's coefficient or Mander's M2 coefficients were calculated using the ImageJ software JACoP plugin with manually set thresholds. Immunoblot and immunoprecipitation experiments were repeated at least three times and representative blots are shown. Data were tested by an unpaired two-tailed t test or one-way ANOVA followed by Dunnett's test or Tukey's multiple comparison test.

## Reporting summary

Further information on research design is available in the Nature Portfolio Reporting Summary linked to this article.

## Data availability

The authors declare that the data supporting the findings of this study are available within the article and its Supplementary Information files. MS proteomics data used in this study is deposited to jPOSTrepo (a

repository that is in the ProteomeXchange consortium) with the dataset identifier JPST002166 [https://repository.jpostdb.org/entry/JPST002166.0] and PXD042419. All other relevant data supporting the findings of this study are available on request. Source data are provided with this paper.

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

## Acknowledgements

This work was supported in part by a Grant-in-Aid for Scientific Research (22H04809, 22KK0096, 22H02868, 21K07023, 21K19376, 19H03471), by The Japan Agency for Medical Research and Development (AMED) (JP22fk0108130h0403), by Takeda Science foundation (to T.N.), by The Chemo-Sero-Therapeutic Research Institute (to T.N. and I.N.), Senri Life Science Foundation (to T.N.), by the Joint Research Project of the Institute of Medical Science, the University of Tokyo (to I.N.), by the Yakult Foundation (to I.N.), by the Research Center for GLOBAL and LOCAL Infectious Diseases, Oita University (2022B08) (to I.N.), by the Grants-in-Aid for Research from the National Center for Global Health and Medicine (22A1016) and by Daiichi Sankyo Foundation of Life Science (to I.N.).

## Author contributions

T.N., H.T., and I.N. conceived the study, T.N. and I.N. provided reagents, and wrote the paper. T.N., H.T., J.I., K.K., A.M.N., J.S., S.I., and K.M. performed the experiments and analyzed the data.

## Competing interests

The authors declare no competing interests.
