## [Peer Review File · Nature Communications]

Rab41-mediated ESCRT machinery repairs membrane rupture by a bacterial toxin in xenophagyReviewer #1 (Remarks to the Author):

The manuscript of Takashi Nozawa et al. focuses on the role of Rab GTPases and ESCRT machinery in xenophagy. The authors developed a siRNA screening towards Rab GTPases and found that knockdown of Rab8A, Rab19, Rab41, and Rab44 affect GAS proliferation through ATG5-dependent autophagy. To characterize the role of Rab41 involved in GAS elimination, the authors suggested Rab41 recruits VPS4 to SLO-damaged GcAVs and facilitates the membrane repair by ESCRT complex. Rab41 localizes to damaged endomembrane independent of its GTPase activity and prenylation. They further identified TOM1L2 by Rab41 immunoprecipitation in response to endomembrane injury. Together, the authors proposed a TOM1L2-Rab41-VPS4 axis that distributes on the damaged compartments/bacteria-containing vacuoles, facilitates ESCRT-mediated membrane repair, and restricts bacterial survival.

This work provides some novel mechanistic insights into endomembrane homeostasis. However, the overall phenotype is inconspicuous, and some conclusions are not convincing. Therefore, the authors have to answer my comments before acceptance.

Major comments:

1. Most of the conclusions were obtained by gene siRNA knockdown throughout the manuscript. Due to the variable efficiency of gene knockdown, it is difficult for the authors to draw definite conclusions. For example, to demonstrate one of the most important findings that Rab41 and TOM1L2, but not ESCRT-III, are required for VPS4 localization to damaged membranes. The authors should generate Rab41 and TOM1L2 knockout cell lines (since Rab41 and TOM1L2 are not essential genes) and examine whether VPS4 is recruited to GcAVs. The data present in Fig.4b and c are not convincing.
2. The TOM1L2-Rab41 and Rab41-VPS4 interactions are essential parts of this manuscript. Mutations of the GTPase activity or C-term of Rab41 did not affect VPS4 recruitment in Fig.4, which indicates these mutations have no effect on the binding of both TOM1L2-Rab41 and Rab41-VPS4 interfaces. The authors need further characterize the critical residues of Rab41 that are required for interaction with TOM1L2 and VPS4, respectively. And then complementation of these mutants in Rab41 KO cells to validate.
3. In Fig. 6d and e, the Rab41-TOM1L2 interaction was not obviously enhanced upon LLOMe stimulation or GAS infection. The authors should provide immunoblot results of the interaction between Rab41 and endogenous TOM1L2 with or without LLOMe or GAS treatment.
4. If I understand correctly, STAND-HS1 blocks the pore-forming activity of SLO. However, xenophagy induction also requires SLO-mediated vacuolar damage. It is impossible to distinguish the membrane damage and xenophagy induction by anti-SLO intrabody. The conclusion in Fig. 3 is suspicious.
5. Does LLOMe-induced VPS4 activation also require Rab41 and TOM1L2? And please discuss the similarities and differences between LLOMe and GAS infection-caused endomembrane injury.
6. Whether Rab41 and TOM1L2 are also involved in other ESCRT-mediated functions? Such as vesicle budding and plasma membrane repair.

Minor comments and errors:

1. The authors proposed that TOM1L2 interacts with Rab41 in response to endomembrane injury. Is the association of Rab41 with VPS4 regulated by membrane damage or constitutive? Are VPS4 and other ESCRT proteins present in the MS results of the GFP-Rab41 interactome?
2. Human has two isoforms of VPS4, VPS4A and VPS4B. The authors did not specify which VPS4 was tested.
3. In lines 118 and 120, the authors mistakenly wrote "Rab44" as "Rab41".
4. Line 48, does ESCRT-0 account for a complex of the ESCRT machinery?
5. Line 204, TSG101 is an ESCRT-I subunit instead of ESCRT-II.
6. Fig. 7d, the authors mistakenly wrote "VPS4" as "Rab41"

Reviewer #2 (Remarks to the Author):

In their manuscript entitled "Rab41-mediated ESCRT 1 machinery repairs membrane rupture by bacterial toxin in xenophagy" the authors describe a new pathway in xenophagy of intracellular

bacteria. They identified Rab41 as a central mediator of protein recruitment to the xenophagolysosomal vacuole. Upon damage of the vacuolar membrane, through the bacterial cytolysin SLO, the increased intracytoplasmic Ca²⁺ concentration would trigger the recruitment of ESCRT-I, II and III machinery. In parallel, an unknown signal would promote the recruitment of TOM1L2 which in turn recruits Rab41 followed by VPS4 which interact with ESCRT-III promoting repair and limiting bacterial proliferation.

This manuscript reports new findings and new concepts that are potentially interesting for researchers in the fields of cellular infection and intracellular trafficking. Besides identifying new proteins involved in the maintenance of vacuolar homeostasis, such as Rab 41 and TOM1L2, the authors claim that Rab41 involvement in this process is independent of its GTPase activity. However, in its current form the manuscript has several flaws that need to be addressed.

Major concerns:

- 1) The manuscript completely lacks any information on the siRNAs used. In addition, the authors do not show whether the siRNAs used are specifically down-regulating the expression of the targeted protein. The authors need to show western blots or qRT-PCR data showing the reduced expression of their targets, before taking any conclusion.
- 2) The authors do not use a siRNA control (such as scrambled siRNA or a siRNA targeting luciferase for example). They use as control the non-transfected cells. This is a problem, as it is not unusual that cells transfected with a control (unrelated) siRNA behave slightly different from non-transfected cells. Given the small effects shown for example in Figure 1A, where the bacterial survival goes at the best from 1 to 1.5, the use of a control siRNA is of major importance.
- 3) The great majority of the data is generated under overexpression of tagged proteins, including microscopy, immunoprecipitation, protein-protein interaction assays. The authors need to make the effort to corroborate their data in conditions of endogenous expression.
- 4) Some data was obtained from the analysis of n>10 vacuoles. This is vague and might not be enough. Were those analyses done in blind? (not mentioned) To how many cells does it correspond?
- 5) The fact that the role of Rab41 in this process is independent of its GTPase activity is surprising. This implies that Rab41 is acting here as an adaptor protein, which is somehow a perturbing concept. Any evidence showing that the Rab41 mutants used are indeed dead for its GTPase activity?

Other concerns:

- 1) Figure 1d shows different numbers of experiments per condition, please explain why.
- 2) Figure 2c is poorly convincing given the overall labelling of SNF8. Its recruitment to the GcAVs is not obvious.
- 3) Figure 1d in the y axis please correct CHMP4B
- 4) Figure 2f, g the differences are really small and the data is obtained from only 10 GcAVs. These data are not strong. Have the authors performed statistical analysis?
- 5) From figure 3 onwards there is a problem in the text references to panel figures. Some panels do not exist and others are incorrect. This needs to be corrected, otherwise it is difficult to follow. As an example, in line 176 the panel d is not showing what is claimed. Panel d needs to be mentioned in line 178 together with panel e. Panel f which is mentioned in line 178 needs to be mentioned in line 180. Line 165 mentions a panel h which does not exist in figure 3.
- 6) Concerning supplementary figure 1, I was wondering if cytolysins produced by *S. aureus* are forming pores large enough to allow Ca²⁺ efflux from the vacuole? If not, what is the signal for the recruitment of CHMP4B?
- 7) In figure 4, the major issue concerns the overexpression of proteins, which is not the best way to assess protein-protein interactions.
- 8) In figure 5a the DAPI staining clearly shows dots outside the cell nucleus, which can be compatible with bacteria. Please explain.
- 9) Figure 6d and 6e, the data is difficult to evaluate and appreciate. The authors need to show the levels of the GFP immunoprecipitated (IP GFP, IB GFP). Only then we can appreciate the levels of FLAG immunoprecipitated and conclude whether FLAG precipitates more or less in a condition than in the other.

Experimental Procedures:

The methods need to be carefully reviewed. In their current form, a lot of information and important details are missing thus preventing the validation or replication of the data by other

labs. The authors need to provide detailed description of the experimental procedures they follow, if this is too extensive they can add it as supplementary material. Below follows a non-exhaustive list of issues raised by the "experimental procedures":

- 1) Gentamicin (50 ug/ml) is added to cells in culture. Why? At such high concentration gentamicin is able to enter into cells. In addition to eliminate extracellular bacteria during infection experiments, the concentration of gentamicin is increased to 100 ug/ml for long time periods (until 6h). This seems excessive. How can the authors ensure that gentamicin is not entering into cells and killing part of the intracellular bacteria?
- 2) Line 440 indicates that several DNA fragments encoding human proteins were obtained by PCR on total mRNA from HeLa or HEK293T cells. The later are not human cells, they are from Hamster. This needs to be corrected. What fragments encode human and non-human proteins.
- 3) The sequences of all the primers and siRNAs used are missing. This needs to be added. More details on siRNAs are required: sequences, how many siRNAs per gene, methods and protocols for transfection, time between transfection and analysis, ...
- 4) Line 485: the authors mentioned "as described previously" ... where? A reference is needed or the full protocol is required.
- 5) The authors provide a brief description of the purification method for the GST tagged protein (lines 491-495). What about the MBP-tagged protein? This protocol is missing. Please add it.
- 6) On fluorescence microscopy the authors should provide the specifications of the camera and objectives used. In addition, they don't mention how the images have been treated? Which software used? Are the images shown projections or single plans?
- 7) Line 555, the authors mentioned Pearson's coefficient. I couldn't find where they use it.

Figure legends:

In general, the Figure legends are poorly written. They need to be improved to provide more details on what they show. As a few examples:

- 1) They should inform whether confocal microscopy images are full projections of different plans or instead they are single plans.
- 2) Data obtained from overexpression or endogenous proteins need to be clearly indicated in the legend.
- 3) Abbreviations in the figure need to be full written in the legend (for example in Figure 4j, what CBB means? Coomassie brilliant Blue? Not clear.)
- 4) The number of experiments and the N for the different panels need to be clearly mentioned. In the way they are presented is difficult to understand. For example, in figure 1d the different conditions presented in the graph do not have the same number of dots. Does this mean that the number of experiments is not the same? Also mentioning that $n > x$ is weird, would be preferable to mention $x < n < y$.
- 5) Also in Figures 1g, 2f, 2j, 5b the authors need to indicate to what corresponds the shadow lines. Is that data dispersion? Need clarification.

Reviewer #3 (Remarks to the Author):

The manuscript by Nozawa et al sheds light on the role of ESCORT machinery and Rab41/ TOMIL2 complex in the process of xenophagy. Through series of experiments carried out using a great variety of microscopic and other biochemical techniques, they provide a detailed, step-by-step description of the process. Overall, the manuscript is well-composed, and the figures are organized and well-presented. Please find my more detailed comments below.

Abstract: the abstract can benefit from inclusion of the names of techniques and details of the experimental method that generated reported findings. For example, the sentence in lines 21-24 can be edited to state: "Confocal microscopy revealed that the ESCORT components were recruited..." and so on. These additions will make the abstract more informative and improve readers' experience with it.

Quantitative methodologies and descriptive statements: there are several statements through the manuscript that can benefit from more specific quantitative comparisons. For example, in line 263 the authors report that the recruitment of Rab41 increased through the time of infection. Can you

be more specific by how much? At what time point? The descriptiveness of this statement can be mitigated by the % comparison or a similar metric. Likewise, the finding in line 190 states that "these signals rarely colocalized." Can this be quantified? "Rarely" appears as a vague descriptor. Many other findings through the manuscript can be better supported by more explicit quantitative comparisons.

IP-MS experiment in Fig. 6A: I found multiple issues with the IP-MS experiment and presentation of its results. First, there appear to be no replicates. IP and AP-MS experiments are notoriously variable, and it is a standard in the field to include 3-4 replicates of each condition. The quantified protein abundances across those are then averaged and some statistical testing or software-based methodology (e.g., SAINT or COMPASS) are employed to distinguish the true interactors from the commonplace contaminants and non-specific binders. To identify interactors of Rab41, the authors have employed some unclear cutoff of FC of ~5-fold between LLOMe and DMSO treated samples. More rigorous approach and statistical testing is needed to establish which proteins are true interactors in this experiment.

Further, GFP alone control was included but not used in any way. The authors should filter out all protein identifications that appear in in experimental samples at the levels comparable to the GFP only control, as these proteins cannot be true interactors of Rab41 and are most certainly non-specific contaminants.

Lastly, the provided heat map of the results is very misleading, as it depicts protein categories across all detected proteins in all sample types. The vast majority of these proteins are not specific to Rab41 or the treatment conditions, and having them highlighted in the figure is confusing and adds no value.

Given the extent of the follow-up experiments, TOM1L2 was likely correctly identified by the IP-MS experiments. However, the rest of the proteins that appear in the figure should be treated with caution and cannot be confidently presumed to be involved with Rab41. The authors should consider redoing the IP-MS experiment including 3-4 replicates and employing more statistically rigorous approach to interactor detection. These finding could be presented as a heat map of filtered interactors with protein classes marked, like in the current Fig 6A, or a simple table. Alternatively, the authors can perform a targeted experiment quantifying abundances of several highlighted proteins across sample types, validating their pilot data presented in Fig. 6A.

Grammatical errors: please revise lines 44-48 and 304-306.

Reviewer #1 (Remarks to the Author):

The manuscript of Takashi Nozawa et al. focuses on the role of Rab GTPases and ESCRT machinery in xenophagy. The authors developed a siRNA screening towards Rab GTPases and found that knockdown of Rab8A, Rab19, Rab41, and Rab44 affect GAS proliferation through ATG5-dependent autophagy. To characterize the role of Rab41 involved in GAS elimination, the authors suggested Rab41 recruits VPS4 to SLO-damaged GcAVs and facilitates the membrane repair by ESCRT complex. Rab41 localizes to damaged endomembrane independent of its GTPase activity and prenylation. They further identified TOM1L2 by Rab41 immunoprecipitation in response to endomembrane injury. Together, the authors proposed a TOM1L2-Rab41-VPS4 axis that distributes on the damaged compartments/bacteria-containing vacuoles, facilitates ESCRT-mediated membrane repair, and restricts bacterial survival. This work provides some novel mechanistic insights into endomembrane homeostasis. However, the overall phenotype is inconspicuous, and some conclusions are not convincing. Therefore, the authors have to answer my comments before acceptance.

We would like to thank the reviewer for providing valuable comments regarding the significance of our study. The reviewer also raised valid and significant concerns, which have helped us tremendously in improving this revised manuscript.

Major comments:

Most of the conclusions were obtained by gene siRNA knockdown throughout the manuscript. Due to the variable efficiency of gene knockdown, it is difficult for the authors to draw definite conclusions. For example, to demonstrate one of the most important findings that Rab41 and TOM1L2, but not ESCRT-III, are required for VPS4 localization to damaged membranes. The authors should generate Rab41 and TOM1L2 knockout cell lines (since Rab41 and TOM1L2 are not essential genes) and examine whether VPS4 is recruited to GcAVs. The data present in Fig.4b and c are not convincing.

We thank the reviewer for constructive advice. As suggested, we have generated *Rab41* and *TOM1L2* knockout cell lines and investigated the involvement of these proteins in VPS4-mediated GcAV homeostasis and host defense against intracellular bacteria. Knockout of *Rab41* significantly reduced the VPS4 signal on GcAVs, and complementation of Rab41 expression rescued VPS4 recruitment (Fig. 4d, e). In addition, the acidification of xenophagolysosomes and suppression of bacterial growth were also diminished by *Rab41*-knockout (Fig. 1h, i, j). Knockout of *TOM1L2* also decreased VPS4 recruitment to xenophagolysosome and impaired bactericidal activity (Fig. 8h, i, j, k). In the revised manuscript, we concluded that Rab41 and TOM1L2 are critical for VPS4 dynamics in xenophagy.

The TOM1L2-Rab41 and Rab41-VPS4 interactions are essential parts of this manuscript. Mutations of the GTPase activity or C-term of Rab41 did not affect VPS4 recruitment in Fig.4, which indicates these mutations have no effect on the binding of both TOM1L2-Rab41 and Rab41-VPS4 interfaces. The authors need further characterize the critical residues of Rab41 that are required for interaction with TOM1L2 and VPS4, respectively. And then complementation of these mutants in Rab41 KO cells to validate.

As suggested by the reviewer, we have tried to identify the critical residues of Rab41 that are important for the interaction with VPS4 and TOM1L2. We successfully identified S94 in Rab41 as a critical amino acid to interact with VPS4 in the revised manuscript (Fig. 4l). We mentioned these results as "*Phosphorylation of Rab GTPase is known as a possible regulatory mechanism, and the phosphorylation sites are located in conserved Rab family/Rab subfamily motif and complementarity determining regions. We next substituted candidate phosphorylation site S94 to A (Alanine) in Rab41 and found that S94A mutation reduced the interaction with endogenous VPS4 (Fig. 4l).*" In addition, exogenic expression of Rab41 S94A mutant in Rab41-knockout cells did not recover bactericidal activity (Fig. 4m), suggesting that S94 residue that is critical for the interaction with VPS4 is required for Rab41-mediated xenophagy against GAS. In future studies, we will examine whether this S94 residue is phosphorylated and identify the responsible kinase, which would reveal the regulatory mechanism of xenophagy in response to bacterial infection.

In Fig. 6d and e, the Rab41-TOM1L2 interaction was not obviously enhanced upon LLOMe stimulation or GAS infection. The authors should provide immunoblot results of the interaction between Rab41 and endogenous TOM1L2 with or without LLOMe or GAS treatment.

We have examined endogenous Rab41-TOM1L2 interaction in the revised manuscript. Our co-immunoprecipitation assay demonstrated that Rab41 interacts with TOM1L2 in response to LLOMe treatment (Fig. 7d), consistent with Rab41-binding protein proteome analysis (Fig. 7a).

If I understand correctly, STAND-HS1 blocks the pore-forming activity of SLO. However, xenophagy induction also requires SLO-mediated vacuolar damage. It is impossible to distinguish the membrane damage and xenophagy induction by anti-SLO intrabody. The conclusion in Fig. 3 is suspicious.

In this experiment, STAND-HS1 is expressed in the cytosol, therefore STAND-HS1 can access cytosolic bacteria but not bacteria inside endosome (outer luminal environment). Hence, STAND-HS1 does not inhibit induction of xenophagy (damage to endosomal membranes), but only membrane damage to xenophagolysosomes (which incorporate cytoplasmic components). This is illustrated in Fig. 3i with an explanation in the revised manuscript.

Does LLOMe-induced VPS4 activation also require Rab41 and TOM1L2? And please discuss the similarities and differences between LLOMe and GAS infection-caused endomembrane injury.

We thank the reviewer for constructive advice. This point is essential to identify the function of Rab41. First, we need to explain the effects of Rab41-knockdown on ESCRT-III (CHMP4B) dynamics during LLOMe-treatment. CHMP4B-positive small vacuoles appeared 10 min after LLOMe treatment and decreased after 60 min in control cells (Fig. 6a, b). By contrast, VPS4A-depleted cells harbored numerous small CHMP4B dots even after 60 min and showed large CHMP4B-positive vacuoles (Fig. 6a, b, c). Importantly, in Rab41 knockdown cells, small CHMP4B puncta reduced from 10 to 60 min after LLOMe treatment, but enlarged CHMP4B-positive vacuoles were accumulated (Fig. 6a, b, c). These enlarged vacuoles were positive for both LC3 and LAMP1, suggesting that Rab41 is mainly involved in the homeostasis of autolysosomes through VPS4 and ESCRT machinery. Indeed, VPS4 was not targeted to LC3-positive enlarged lysosomes (autolysosomes) in *Rab41*-knockout cells (Fig. 6e). This theory is consistent with the data of kinetics of Rab41 recruitment to lysosomes, in which endogenous Rab41 is distributed to lysosomes after LC3 and LGALS3 recruitment (Fig. 5m). Even in case of GAS infection, recruitment of Rab41 to around bacteria was abolished by ATG7-knockout, indicating that Rab41 targets xenophagic membrane. So, why should Rab41 selectively function in autolysosomal homeostasis? The reasons for this are not yet clear, but we have discussed in the Discussion section of this revised manuscript as "*However, both exogenous and endogenous EmGFP-Rab41 targeted only LC3-positive bacteria-containing vacuoles, and knockout of ATG7 abolished Rab41 recruitment to GAS. Therefore, it is likely that Rab41 functions in the homeostasis of autophagic lysosomal vacuoles. Why is Rab41 not involved in recruiting VPS4 to lysosomes and its repair but only needed for xenophagolysosomes repair? The endosomal membrane damage induces xenophagy (lysophagy), which can be eliminated, but for the membrane damage of the last-resort xenophagolysosome, another immune system should be activated to protect the cells. Of note, RNF5 and RNF185 were identified to interact with Rab41 upon lysosomal membrane damage (Fig. 7a). Because RNF5 and RNF185 have been reported to facilitate various innate immune responses to pathogens, it would be informative to reveal how Rab41 associates with these E3 ligases and their contribution to host defense systems.*".

Whether Rab41 and TOM1L2 are also involved in other ESCRT-mediated functions? Such as vesicle budding and plasma membrane repair.

We have not examined the involvement of Rab41 and TOM1L2 in other ESCRT-mediated functions yet, but we are also interested in this point and have plan to investigate it in future studies.

Minor comments and errors:

The authors proposed that TOM1L2 interacts with Rab41 in response to endomembrane injury. Is the association of Rab41 with VPS4 regulated by membrane damage or constitutive? Are VPS4 and other ESCRT proteins present in the MS results of the GFP-Rab41 interactome?

We examined endogenous Rab41-VPS4 interaction and found that Rab41 binds to VPS4 in both non-infected and infected conditions (Fig. 4g). VPS4A was also detected as Rab41-binding protein in Fig. 7A, but VPS4 was not significantly enriched in LLOMe-treated cells. These results are also consistent with the results of endogenous Rab41-VPS4 interaction.

Human has two isoforms of VPS4, VPS4A and VPS4B. The authors did not specify which VPS4 was tested.

We have examined the interaction between Rab41 and VPS4A or VPS4B, and showed that Rab41 can interact with both VPS4A and VPS4B (Fig. S6a).

In lines 118 and 120, the authors mistakenly wrote “Rab44” as “Rab41”.

As suggested, we modified “Rab41” to “Rab44” in the revised manuscript.

Line 48, does ESCRT-0 account for a complex of the ESCRT machinery?

We have revised this sentence as follow " *The ESCRT machinery can be divided into three functionally distinct complexes known as ESCRT- 0, ESCRT- I, ESCRT-II and ESCRT-III. ESCRT-III that is consist of α -helical CHMP proteins is recruited by ESCRT-II and critical for driving membrane constriction through the formation of membrane-binding spirals that mediate membrane deformation and scission, in cooperation with the ATPase VPS4.*" in the revised manuscript.

Line 204, TSG101 is an ESCRT-I subunit instead of ESCRT-II.

We have modified to "ESCRT-I" in this revised manuscript.

Fig. 7d, the authors mistakenly wrote “VPS4” as “Rab41”

As suggested we modified to VPS4.

Reviewer #2 (Remarks to the Author):

In their manuscript entitled “Rab41-mediated ESCRT 1 machinery repairs membrane rupture by bacterial toxin in xenophagy” the authors describe a new pathway in xenophagy of intracellular bacteria. They identified Rab41 as a central mediator of protein recruitment to the

xenophagolysosomal vacuole. Upon damage of the vacuolar membrane, through the bacterial cytolysin SLO, the increased intracytoplasmic Ca²⁺ concentration would trigger the recruitment of ESCRT-I, II and III machinery. In parallel, an unknown signal would promote the recruitment of TOM1L2 which in turn recruits Rab41 followed by VPS4 which interact with ESCRT-III promoting repair and limiting bacterial proliferation.

This manuscript reports new findings and new concepts that are potentially interesting for researchers in the fields of cellular infection and intracellular trafficking. Besides identifying new proteins involved in the maintenance of vacuolar homeostasis, such as Rab 41 and TOM1L2, the authors claim that Rab41 involvement in this process is independent of its GTPase activity. However, in its current form the manuscript has several flaws that need to be addressed.

We appreciate the reviewer for bringing up valid and significant concerns, which have greatly assisted us in enhancing this revised manuscript.

Major concerns:

The manuscript completely lacks any information on the siRNAs used. In addition, the authors do not show whether the siRNAs used are specifically down-regulating the expression of the targeted protein. The authors need to show western blots or qRT-PCR data showing the reduced expression of their targets, before taking any conclusion.

We have added all information of siRNAs (ID numbers) in Experimental Procedure section of this revised manuscript. We also showed the effects of siRNA knockdown on the expressions of target proteins by real-time quantitative PCR (RT-qPCR) or immunoblot (Supplementary Fig. 1a, 3e and).

The authors do not use a siRNA control (such as scrambled siRNA or a siRNA targeting luciferase for example). They use as control the non-transfected cells. This is a problem, as it is not unusual that cells transfected with a control (unrelated) siRNA behave slightly different from non-transfected cells. Given the small effects shown for example in Figure 1A, where the bacterial survival goes at the best from 1 to 1.5, the use of a control siRNA is of major importance.

We agree with that negative control using siRNA control is critical in these experiments. In our experiment, we used non-targeting siRNA (siNT, 12935300 or 4390843, Thermo Fisher Scientific) as a negative control and quantified the effects of Rab knockdown compared to siRNA control cells.

The great majority of the data is generated under overexpression of tagged proteins, including

microscopy, immunoprecipitation, protein-protein interaction assays. The authors need to make the effort to corroborate their data in conditions of endogenous expression.

We thank the reviewer for constructive advices. As suggested, we have examined endogenous protein-protein interactions (Rab41-VPS4 and Rab41-TOM1L2). Our endogenous co-immunoprecipitation demonstrated that Rab41 interacts with both VPS4 and TOM1L2 (Fig. 4g and 7d), and also showed that Rab41-TOM1L2 interaction is triggered upon LLOMe treatment (Fig. 7d). Moreover, to observe endogenous localization of Rab41, we have generated GFP knock-in (EmGFP-Rab41) HeLa cell lines using CRISPR/Cas9 genome editing and TrueTag donor DNA (Thermo Fisher Scientific) (Supplementary Fig. 7a, b). Endogenous EmGFP-Rab41 (eEmGFP-Rab41) clearly colocalized with CHMP4B-positive bacteria (Fig. 5c) and redistributed to autolysosomal compartment in response to membrane damage (Fig. 5k, l, m).

Some data was obtained from the analysis of n>10 vacuoles. This is vague and might not be enough. Were those analysis done in blind? (not mentioned) To how many cells does it corresponds?

As suggested by the reviewer, we have re-analyzed and increased n number to validate previous results. For example, in an experiment to quantify CHMP4B recruitment, we did not look at the CHMP4B signal, but took random pictures and quantify the CHMP4B and LC3 signals by image analysis. In case we analyzed 20 vacuoles, which corresponds to about 60~100 cells.

The fact that the role of Rab41 in this process is independent of its GTPase activity is surprising. This implies that Rab41 is acting here as an adaptor protein, which is somehow a perturbing concept. Any evidence showing that the Rab41 mutants used are indeed dead for its GTPase activity?

Rab41 mutants used in this study are T44N and Q89L mutants of Rab41. T44 and Q89 residues are located in conserved motif among Rab GTPase proteins well-known to be important for hydrolysis of GTP to GDP and exchange from GDP to GTP, respectively. Indeed, Rab41 T44N is reported to inhibit Rab41-mediated ER-to-Golgi trafficking (PMID: 23936529), indicating that Rab41 has role in transport pathway in GTPase-dependent mechanism. We then investigated whether Rab41 mutants used in this study have GTPase activity or are dead for the activity using RUSH (Retention Using Selective Hooks) assay. We observed localization of reporter (α -Mannosidase II-SBP-mCherry), which is trafficked from the ER to the Golgi in the presence of biotin, and quantified the reporter signals reached to the Golgi (GM130) after treatment of biotin for 1 h. Overexpression of wild-type and Q89L Rab41 did not affect the trafficking, but Rab41 T44N significantly reduced the transport of the reporter from the ER to the Golgi (Supplementary Fig. 4a, b), demonstrating that Rab41 T44N expression diminished

GTPase-dependent function of Rab41. Whereas, expression of Rab41 T44N rescued the recruitment of VPS4 to GcAVs in *Rab41* KO cells (Fig. 4d, e) and inhibited bacterial proliferation (Fig. 4f) at same content with wild-type Rab41. Therefore, it was demonstrated that Rab41 functions as an adaptor protein in GTPase-independent manner.

Other concerns:

Figure 1d shows different numbers of experiments per condition, please explain why.

This analysis was performed by several researchers in blind, and one did only one or two siRab samples. This is the reason why number of experiments differ among Rabs, however, there is no problem in statistical significance.

Figure 2c is poorly convincing given the overall labelling of SNF8. Its recruitment to the GcAVs is not obvious.

We have changed to another more representative images (Fig. 2c).

Figure 1d in the y axis please correct CHMP4B

We modified spell of CHMP4B in y axis (Fig. 2d).

Figure 2f, g the differences are really small and the data is obtained from only 10 GcAVs. These data are not strong. Have the authors performed statistical analysis?

In this revised experiment, this experiment was repeated to increase the number of GcAVs analyzed (< 30 vacuoles in each condition). We performed this experiment to understand the kinetics of the recruitment of ESCRT-III to autophagic membrane. Therefore, we did not compare the recruitment of CHMP4B and LAMP1 to LC3.

From figure 3 onwards there is a problem in the text references to panel figures. Some panels do not exist and other are incorrect. This needs to be corrected, otherwise it is difficult to follow.

As an example, in line 176 the panel d is not showing what is claimed. Panel d needs to be mentioned in line 178 together with panel e. Panel f which is mentioned in line 178 needs to be mentioned in line 180. Line 165 mentions a panel h which does not exist in figure 3.

We thank the reviewer for suggestions. We have carefully modified the panel figures and the citation in the revised manuscript.

Concerning supplementary figure 1, I was wondering if cytolysins produced by *S. aureus* are forming pores large enough to allow Ca²⁺ efflux from the vacuole? If not, what is the signal for the recruitment of CHMP4B?

S. aureus produces cytolysins such as α -toxin and to form pores large enough to induce calcium mobilization.

In figure 4, the major issue concerns the overexpression of proteins, which is not the best way to assess protein-protein interactions.

We have performed endogenous protein-protein co-immunoprecipitation analysis and showed that endogenous Rab41 interacts with VPS4 (Fig. 4g).

In figure 5a the DAPI staining clearly shows dots outside the cell nucleus, which can be compatible with bacteria. Please explain.

These DAPI-staining dots were frequently observed when we transiently transfected plasmids. Thus, it is likely to be an aggregate of plasmids.

Figure 6d and 6e, the data is difficult to evaluate and appreciate. The authors need to show the levels of the GFP immunoprecipitated (IP GFP, IB GFP). Only then we can appreciate the levels of FLAG immunoprecipitated and conclude whether FLAG precipitates more or less in a condition than in the other.

In this revised manuscript, we have examined the interaction of endogenous Rab41 and TOM1L2 and showed that TOM1L2 was precipitated with endogenous Rab41 only when cells were treated with LLOMe (Fig. 7d).

Experimental Procedures:

The methods need to be carefully reviewed. In their current form, a lot of information and important details are missing thus preventing the validation or replication of the data by other labs. The authors need to provide a detailed description of the experimental procedures they follow, if this is too extensive they can add it as supplementary material. Below follows a non-exhaustive list of issues raised by the “experimental procedures”:

We thank for your suggestion. We have fully revised the Experimental Procedure Section in the revised manuscript.

Gentamicin (50 μ g/ml) is added to cells in culture. Why? At such a high concentration gentamicin is able to enter into cells. In addition to eliminating extracellular bacteria during infection experiments, the concentration of gentamicin is increased to 100 μ g/ml for long time periods (until 6h). This seems excessive. How can the authors ensure that gentamicin is not entering into cells and killing part of the intracellular bacteria?

Cell lines are frequently maintained medium with streptomycin and penicillin to inhibit microbial contaminations, but GAS JRS4 strain used in this study is resistant to streptomycin. We then used gentamicin instead of streptomycin and penicillin. As you point out, it has been noted that some cell types and treatment concentrations can penetrate into cells, so it is important to set up carefully according to the experimental conditions (PMID: 29312891). The concentrations of gentamicin used in this study are frequently (in more than hundreds of papers) used in "Gentamicin protection assay" to completely kill the extracellular bacteria in HeLa cells. GAS strains including JRS4 used in this study is sensitive to gentamicin, but the number of survival bacteria increase with time in HeLa cells (Fig. A). Moreover, although knockout of ATG5 increased intracellular bacteria (Fig. A), increase of gentamicin concentration to 200 ug/ml did not change the survival bacteria at all (Fig. B). Therefore, of course we cannot exclude the possibility that small part of intracellular bacteria are killed by gentamicin, but we do not believe that this concentration of gentamicin has an effect sufficient to overturn the experimental results.

Line 440 indicates that several DNA fragments encoding human proteins were obtained by PCR on total mRNA from HeLa or HEK293T cells. The later are not human cells, they are from Hamster. This needs to be corrected. What fragments encode human and non-human proteins. As suggested, we separately described the mRNA from HeLa or HEK293T. HEK293T is a cell line of human embryonic kidney.

The sequences of all the primers and siRNAs used are missing. This needs to be added. More details on siRNAs are required: sequences, how many siRNAs per gene, methods and protocols for transfection, time between transfection and analysis, ...

We have added siRNA information in Experimental procedure in this revised manuscript. We also revised as follow " For knockdown experiments, HeLa cells were seeded at 4×10^4 cell/well in 24-well plate and 10 pmol siRNA oligonucleotides Rab1A (s229381; Thermo Fisher Scientific), Rab1B (s119; Thermo Fisher Scientific), Rab2A (s11660; Thermo Fisher Scientific), Rab2B (s39689; Thermo Fisher Scientific), Rab3A (s11668; Thermo Fisher

Scientific), Rab3B (s11671; Thermo Fisher Scientific), Rab3C (s41884; Thermo Fisher Scientific), Rab3D (s18327; Thermo Fisher Scientific), Rab4A (s11675; Thermo Fisher Scientific), Rab4B (s28802; Thermo Fisher Scientific), Rab5A (s11679; Thermo Fisher Scientific), Rab5B (s11681; Thermo Fisher Scientific), Rab5C (s11710; Thermo Fisher Scientific), Rab6A (s11685; Thermo Fisher Scientific), Rab6B (s28326; Thermo Fisher Scientific), Rab7A (s15444; Thermo Fisher Scientific), Rab7B (s50335; Thermo Fisher Scientific), Rab8A (s8681; Thermo Fisher Scientific), Rab8B (s28634; Thermo Fisher Scientific), Rab9A (s17916; Thermo Fisher Scientific), Rab9B (s27693; Thermo Fisher Scientific), Rab10 (s21391; Thermo Fisher Scientific), Rab11A (s16703; Thermo Fisher Scientific), Rab11B (s17649; Thermo Fisher Scientific), Rab12 (s47369; Thermo Fisher Scientific), Rab13 (s11692; Thermo Fisher Scientific), Rab14 (s28311; Thermo Fisher Scientific), Rab15 (s51763; Thermo Fisher Scientific), Rab17 (s34609; Thermo Fisher Scientific), Rab18 (s22705; Thermo Fisher Scientific), Rab19 (s53589; Thermo Fisher Scientific), Rab20 (s31158; Thermo Fisher Scientific), Rab21 (s22823; Thermo Fisher Scientific), Rab22A (s32994; Thermo Fisher Scientific), Rab23 (s28568; Thermo Fisher Scientific), Rab24 (s28804; Thermo Fisher Scientific), Rab25 (s32702; Thermo Fisher Scientific), Rab26 (s24590; Thermo Fisher Scientific), Rab27A (s11693; Thermo Fisher Scientific), Rab27B (s11696; Thermo Fisher Scientific), Rab28 (s17912; Thermo Fisher Scientific), Rab29 (s17083; Thermo Fisher Scientific), Rab30 (s26138; Thermo Fisher Scientific), Rab31 (s21731; Thermo Fisher Scientific), Rab32 (s21619; Thermo Fisher Scientific), Rab33A (s17909; Thermo Fisher Scientific), Rab33B (s37939; Thermo Fisher Scientific), Rab34 (s38246; Thermo Fisher Scientific), Rab35 (s21708; Thermo Fisher Scientific), Rab36 (s18462; Thermo Fisher Scientific), Rab37 (s223876; Thermo Fisher Scientific), Rab38 (s24321; Thermo Fisher Scientific), Rab39A (s42009; Thermo Fisher Scientific), Rab39B (s44481; Thermo Fisher Scientific), Rab40A (s21588; Thermo Fisher Scientific), Rab40B (s21588; Thermo Fisher Scientific), Rab40C (s33730; Thermo Fisher Scientific), Rab41 (s51269; Thermo Fisher Scientific), Rab42 (s41783; Thermo Fisher Scientific), Rab43 (s50453; Thermo Fisher Scientific), Rab44 (s53538; Thermo Fisher Scientific), RASEF (s46039; Thermo Fisher Scientific), PDCD6IP (s19465; Thermo Fisher Scientific), TSG101 (s14439; Thermo Fisher Scientific), VPS28 (s27577; Thermo Fisher Scientific), VPS37A (s44037; Thermo Fisher Scientific), VPS25 (s38895; Thermo Fisher Scientific), SNF8 (s22247; Thermo Fisher Scientific), VPS36 (s27276; Thermo Fisher Scientific), CHMP1A (s533441; Thermo Fisher Scientific), CHMP2A (s26027; Thermo Fisher Scientific), CHMP3 (s28474; Thermo Fisher Scientific), CHMP4A (s26441; Thermo Fisher Scientific), CHMP5 (s28237; Thermo Fisher Scientific), CHMP6 (s35990; Thermo Fisher Scientific), CHMP7 (s40780; Thermo Fisher Scientific), VPS4A (s25966; Thermo Fisher

Scientific), VPS4B (s18273; Thermo Fisher Scientific) or non-targeting siRNAs (4390843, 12935300, Thermo Fisher Scientific) was transfected using Lipofectamine 3000 (Invitrogen) 24 h later. Forty eight hour after transfection, cells were used for experiments. " in the revised manuscript.

Line 485: the authors mentioned "as described previously" ... where? A reference is needed or the full protocol is required.

We have added full protocol as follow " **Immunoblot analysis** Cell lysate samples were mixed with equal volume of 2x Laemmli SDS sample buffer and boiled for 5 min. Samples were separated by SDS-PAGE and then transferred to polyvinylidene difluoride (PVDF) membrane. Membrane was blocked with blocking buffer (5% skim milk or Blocking One (Nacalai Tesque)), and incubated with primary antibody diluted with blocking buffer at 4 °C overnight. For immunoblot analysis, the following antibodies were used as a primary antibody: streptolysin O (SLO) (64-001, BioAcademia, 1:1000), HA (HA124; Nacalai Tesque, 1:1000), FLAG (M2; A2220; Sigma-Aldrich, 1:1000), GFP (GF200; 04363-24; Nacalai Tesque, 1:1000), MBP (E8032, New England BioLabs, 1:1000), Rab41 (NBP2-83434; NOVUS Biologicals), VPS4A/B (17673-1-AP, Proteintech, 1:1000), TOM1L2 (ab121716; Abcam; 1:1000), and TOM1L2 (GTX106295; GENETEX, 1:1000). After wash with PBS containing 0.1% Tween-20 three times, membrane was incubated with secondary antibody for 2 h at room temperature. Horseradish peroxidase-conjugated anti-rabbit IgG (Jackson Laboratories), anti-mouse IgG (Jackson Laboratories), and anti-rabbit IgG (conformation-specific; L27A9; 5127; Cell Signaling Technology) were used as secondary antibodies for immunoblots. After wash three times, the membrane was reacted with Chemi-Lumi One Super (Nacalai Tesque), and images were obtained using LAS-4000 mini Luminescent image analyzer (Fujifilm). " in the revised manuscript.

The authors provide a brief description of the purification method for the GST tagged protein (lines 491-495). What about the MBP-tagged protein? This protocol is missing. Please add it.

We have revised this section as follow " **Protein expression and pull-down assay** GST-tagged proteins constructed in pGEX-6P-1 (GE Healthcare Life Sciences) and MBP-tagged proteins in pMAL- c5x (New England Biolabs) were transformed into *Escherichia coli* BL21 (DE3) cells, which were then cultured at 37°C in LB medium supplemented with 100 µg/mL ampicillin, and induced for 3 h at 37°C with 0.3 mM isopropyl β-D-thiogalactopyranoside (Nacalai Tesque). Cells were harvested by centrifugation, washed with PBS, lysed in lysis buffer (40 mM Tris-HCl pH 7.5, 5 mM EDTA, and 0.5 % Triton X-100), sonicated, and cleared by centrifugation. The resulting supernatant of GST tagged protein was incubated with Glutathione Sepharose 4 Fast Flow (GE Healthcare Life Sciences) for 2 h at 4°C. After several washes with lysis buffer, and beads were mixed and reacted with the supernatant of MBP tagged proteins for 2 h at at 4°C . After wash five

times with lysis buffer, the resulting samples were subjected to SDS-PAGE, and followed by coomassie brilliant blue (CBB) staining or immunoblot." in the revised manuscript.

On fluorescence microscopy the authors should provide the specifications of the camera and objectives used. In addition, they don't mention how the images have been treated? Which software used? Are the images shown projections or single plans?

We have revised as follow " Confocal fluorescence micrographs were single slice acquired with an LSM900 laser-scanning microscope with a Plan-Apochromat 63×/1.4 oil DIC objective lens and ZEN software (Zeiss) or FV1000 laser-scanning microscope with an UPlanSApo 100× oil/1.40 objective lens and Fluoview software (Olympus)."

Line 555, the authors mentioned Pearson's coefficient. I couldn't find where they use it. Pearson's coefficient was used in Fig. 5h.

Figure legends:

In general, the Figure legends are poorly written. They need to be improved to provide more details on what they show. As a few examples:

We have revised the figure legends throughout and carefully added detailed explanations.

They should inform whether confocal microscopy images are full projections of different plans or instead they are single plans.

Confocal microscopy images are all single slice images. We have added this information in all figure legends.

Data obtained from overexpression or endogenous proteins need to be clearly indicated in the legend.

We described these information as follow in case that we overexpress GFP-LC3 and observed endogenous CHMP6, " HeLa cells stably expressing GFP-LC3 were infected with GAS for 4 h, fixed, then immunostained for endogenous CHMP6. Cellular and bacterial DNA were stained with DAPI." in the revised manuscript.

Abbreviations in the figure need to be full written in the legend (for example in Figure 4j, what CBB means? Coomassie brilliant Blue? Not clear.)

We added full spelling of the abbreviation throughout the all figure legends.

The number of experiments and the N for the different panels need to be clearly mentioned. In

the way they are presented is difficult to understand. For example, in figure 1d the different conditions presented in the graph do not have the same number of dots. Does this mean that the number of experiments is not the same? Also mentioning that $n > x$ is weird, would be preferable to mention $x < n < y$.

We have revised the way to mention the number of experiments in each data to be more understandable.

Also in Figures 1g, 2f, 2j,5b the authors need to indicate to what corresponds the shadow lines. Is that data dispersion? Need clarification.

We described the explanation of the shadow lines in the revised manuscript.

Reviewer #3 (Remarks to the Author):

The manuscript by Nozawa et al sheds light on the role of ESCORT machinery and Rab41/TOMIL2 complex in the process of xenophagy. Through series of experiments carried out using a great variety of microscopic and other biochemical techniques, they provide a detailed, step-by-step description of the process. Overall, the manuscript is well-composed, and the figures are organized and well-presented. Please find my more detailed comments below.

Abstract: the abstract can benefit from inclusion of the names of techniques and details of the experimental method that generated reported findings. For example, the sentence in lines 21-24 can be edited to state: "Confocal microscopy revealed that the ESCORT components were recruited..." and so on. These additions will make the abstract more informative and improve readers' experience with it.

We appreciate for your positive comments and also thank for your informative advices.

Quantitative methodologies and descriptive statements: there are several statements through the manuscript that can benefit from more specific quantitative comparisons. For example, in line 263 the authors report that the recruitment of Rab41 increased through the time of infection. Can you be more specific by how much? At what time point? The descriptiveness of this statement can be mitigated by the % comparison or a similar metric. Likewise, the finding in line 190 states that "these signals rarely colocalized." Can this be quantified? "Rarely" appears as a vague descriptor. Many other findings through the manuscript can be better supported by more explicit quantitative comparisons.

We thanks for your suggestion. We have revised several description in Results section using quantitative comparisons. For example, we modified the suggested point as follow "

Furthermore, the kinetics revealed that about only less than 5% of GcAVs were Rab41-positive

at 2 hpi but Rab41-positive GcAVs at 5 hpi or later reached 40 % (Fig. 5b). Because ESCRT-III is recruited to GcAVs from 2 to 4 hpi (Fig. 5b), it was suggested that the targeting of Rab41 to GcAVs occurred later than ESCRT-III recruitment." We also added quantification data in Supplementary Fig. 2a.

IP-MS experiment in Fig. 6A: I found multiple issues with the IP-MS experiment and presentation of its results. First, there appear to be no replicates. IP and AP-MS experiments are notoriously variable, and it is a standard in the field to include 3-4 replicates of each condition. The quantified protein abundances across those are then averaged and some statistical testing or software-based methodology (e.g., SAINT or COMPASS) are employed to distinguish the true interactors from the commonplace contaminants and non-specific binders. To identify interactors of Rab41, the authors have employed some unclear cutoff of FC of ~5-fold between LLOMe and DMSO treated samples. More rigorous approach and statistical testing is needed to establish which proteins are true interactors in this experiment.

Further, GFP alone control was included but not used in any way. The authors should filter out all protein identifications that appear in in experimental samples at the levels comparable to the GFP only control, as these proteins cannot be true interactors of Rab41 and are most certainly non-specific contaminants.

Lastly, the provided heat map of the results is very misleading, as it depicts protein categories across all detected proteins in all sample types. The vast majority of these proteins are not specific to Rab41 or the treatment conditions, and having them highlighted in the figure is confusing and adds no value.

Given the extent of the follow-up experiments, TOM1L2 was likely correctly identified by the IP-MS experiments. However, the rest of the proteins that appear in the figure should be treated with caution and cannot be confidently presumed to be involved with Rab41. The authors should consider redoing the IP-MS experiment including 3-4 replicates and employing more statistically rigorous approach to interactor detection. These finding could be presented as a heat map of filtered interactors with protein classes marked, like in the current Fig 6A, or a simple table. Alternatively, the authors can perform a targeted experiment quantifying abundances of several highlighted proteins across sample types, validating their pilot data presented in Fig. 6A.

We appreciate for your valuable comments and totally agree with you on the IP-MS experimental data you pointed out.

As described in Experimental Procedure section of the revised manuscript, we re-performed IP-MS experiment with replicates (n=4 for each condition), and the acquired data were analyzed by ProteinPilot and Progenesis QI. The proteins identified in the GFP-only condition were excluded. In addition, among proteins commonly identified in three conditions (GFP only, DMSO treatment,

and LLOMe treatment), identified proteins showing the highest abundance in GFP only condition were removed from the list (Supplementary Data 1) and subsequent data analysis. Then, the log₂-transformed fold change and the enrichment score (-log₁₀[Pvalue]) in each protein between DMSO and LLOMe treatment conditions were shown in the Volcano plots (revised Fig. 7a).

Grammatical errors: please revise lines 44-48 and 304-306.

We have revised line 44-48 as follow "ESCRT proteins were originally identified as regulators of ubiquitinated cargo sorting into multivesicular bodies but now have been established to have various intracellular membrane dynamics process such as budding of enveloped viruses, vesicle budding, cytokinetic abscission, nuclear envelope maintenance, repair of plasma membrane and endolysosomal membrane, and autophagy." in the revised manuscript. We also revised line 204-306 as follow " Rab41 targets to damaged auto-/xenophago-lysosomes independently of its nucleotide-bound form and lipidation, we then sought to identify the protein that recruits Rab41 to damaged membranes."

Reviewer #1 (Remarks to the Author):

The revised article addressed the majority of the inquiries I had raised. The logical flow and overall quality have shown significant enhancements. With the exception of the mention that there should be four complexes instead of three in line 49, I don't have any additional inquiries.

Reviewer #2 (Remarks to the Author):

The proposed revised version of the manuscript is considerably improved. The authors have made a real effort to address the reviewer's comments by providing new data, clarifying previous data and by improving the text.

In particular, the authors performed new experiments and provide new data: siRNA data were complemented with KO cells, control siRNA was added and interactions were confirmed with endogenous levels of the proteins. The new data fully support the conclusions.

Also, the Figure Legends and Material and Methods sections were revised and improved.

Overall, this manuscript reports important and novel data that will certainly impact the field of plasma membrane repair and bacterial infection.

Reviewer #3 (Remarks to the Author):

The resubmitted manuscript is greatly improved and augmented by additional methodological details and experiments. The redone IP-MS experiments are now up to the standard in the field and instill more confidence in their findings. The authors should add a one sentence summary of the new IP-MS methodology (# of replicates, statistical testing used, MS method, etc.) to the main text of the manuscript and the figure legend (at least N and stat test), so the readers do not need to search through the methods for this important information. With that, my concerns are resolved.

RESPONSE to REVIEWERS' COMMENTS

Reviewer #1 (Remarks to the Author):

The revised article addressed the majority of the inquiries I had raised. The logical flow and overall quality have shown significant enhancements. With the exception of the mention that there should be four complexes instead of three in line 49, I don't have any additional inquiries.

Response) We appreciate for your suggestion. We have modified "three complexes" to "four complexes" in the revised manuscript.

Reviewer #3 (Remarks to the Author):

The resubmitted manuscript is greatly improved and augmented by additional methodological details and experiments. The redone IP-MS experiments are now up to the standard in the field and instill more confidence in their findings. The authors should add a one sentence summary of the new IP-MS methodology (# of replicates, statistical testing used, MS method, etc.) to the main text of the manuscript and the figure legend (at least N and stat test), so the readers do not need to search through the methods for this important information. With that, my concerns are resolved.

Response) As suggested, we have added sentences to explain methodological details of IP-MS experiment in main text and figure legend in the revised manuscript. Main text described as follow "We performed IP-mass spectrometry (IP-MS) analysis of GFP-Rab41 protein complexes in cells stimulated with LLOMe (Fig. 7a). HEK293T cells expressing GFP or GFP-Rab41 were treated with LLOMe and GFP or GFP-Rab41 was precipitated with GFP-Trap beads, and precipitated proteins were identified by mass spectrometry (n=4). The proteins identified in the GFP-only condition or showing the highest abundance in GFP only condition were removed from subsequent data analysis." We also added information of number of biological replicates (n=4) and statistic test (Unpaired two-tailed t test) in figure legend (Fig. 7a).